# Procedural Fairness Through Decoupling Objectionable Data Generating Components

**Zeyu Tang**[1], **Jialu Wang**[2], **Yang Liu**[2,4], **Peter Spirtes**[1], and **Kun Zhang**[1,3]

[1]Department of Philosophy, Carnegie Mellon University
[2]Computer Science and Engineering Department, University of California, Santa Cruz
[3]Machine Learning Department, Mohamed bin Zayed University of Artificial Intelligence
[4]ByteDance Research
`zeyutang@cmu.edu`, {`faldict, yangliu`}`@ucsc.edu`, {`ps7z@andrew., kunz1@`}`cmu.edu`

## Abstract

We reveal and address the frequently overlooked yet important issue of *disguised procedural unfairness*, namely, the potentially inadvertent alterations on the behavior of neutral (i.e., not problematic) aspects of data generating process, and/or the lack of procedural assurance of the greatest benefit of the least advantaged individuals. Inspired by John Rawls's advocacy for *pure procedural justice* (Rawls, 1971; 2001), we view automated decision-making as a microcosm of social institutions, and consider how the data generating process itself can satisfy the requirements of procedural fairness. We propose a framework that decouples the objectionable data generating components from the neutral ones by utilizing reference points and the associated value instantiation rule. Our findings highlight the necessity of preventing *disguised procedural unfairness*, drawing attention not only to the objectionable data generating components that we aim to mitigate, but also more importantly, to the neutral components that we intend to keep unaffected.

## 1 Introduction

The algorithmic fairness literature has presented various notions to capture fairness with respect to the prediction or the prediction-based decision-making (Dwork et al., 2012; Hardt et al., 2016b; Chouldechova, 2017; Zafar et al., 2017), and also notions that are based on causal modeling of the data generating process (Kusner et al., 2017; Kilbertus et al., 2017; Nabi & Shpitser, 2018; Chiappa, 2019; Wu et al., 2019; Coston et al., 2020). Recent survey papers have presented overviews on various (instantaneous) fairness notions (Loftus et al., 2018; Makhlouf et al., 2020; Mehrabi et al., 2021), the fairness consideration in dynamic settings (Zhang & Liu, 2021), as well as the reflection on the connection between algorithmic fairness and the literature from other disciplines, especially moral and political philosophy (Tang et al., 2023b).

In this paper, we denote "procedural fairness" as the fairness considerations specifically pertaining to the data generation process itself. The literature has presented proposals to utilize causal modeling to infuse more procedural emphasis on algorithmic fairness, e.g., path-specific interventional causal effect (Kilbertus et al., 2017; Nabi & Shpitser, 2018; Nabi et al., 2019; 2022), counterfactual causal effect (Kusner et al., 2017), path-specific counterfactual causal effect (Chiappa, 2019; Wu et al., 2019), and different types of predictive rates (Zhang & Bareinboim, 2018b; Coston et al., 2020; Nilforoshan et al., 2022). Previous causal fairness notions focus on providing quantification tools to estimate or bound discrimination in terms of causal effects from protected features on the outcome. In turn, the intended procedural emphasis on process gravitates towards the substantive emphasis on outcome, and the properties of the data generating process itself, according to which the prediction/decision is derived, are under-characterized.[1] The requirements of procedural fairness are violated without noticing, and consequently, the issue of *disguised procedural unfairness* is often overlooked.

Given a causal graph that represents the underlying data generating process, one can follow the graph and perform an inference task for prediction or decision-making. The discrimination within a data

---

[1]Due to space limit, we present the detailed literature review in Appendix A, and discuss differences and connections between our framework and previous works in Appendix B.

generating process, however, can manifest in various forms and originate from different locations, not all of which are closely connected to the protected feature or final outcome. For example, an individual may encounter discriminations based on various aspects of the physical appearance, like a shorter stature (Judge & Cable, 2004; Persico et al., 2004) or a larger build (Puhl & Heuer, 2010). Individuals may also face discriminations due to the linguistic characteristics of their speech and writing (Baugh, 2005) or their accent (Barrett et al., 2022). A household may endure discriminations because of marital or family status (Bruin & Cook, 1997; Lauster & Easterbrook, 2011; Joslin, 2015). Therefore, we use the term "objectionable component" to refer to any problematic aspects in the data generating process, which may include an edge in the local causal module, or segments of a path whose starting and ending nodes are not limited to protected features and final outcome. Data generating components that are not objectionable are accordingly termed "neutral components."

Our goal is to achieve procedural fairness by *decoupling* objectionable components from the data generating process, and making predictions *only* based on the neutral components. We present a framework consisting of reference points and the corresponding value instantiation rule to fulfill the requirements of procedure fairness. Our contributions can be summarized as follows:

- We reveal the frequently overlooked issue of *disguised procedural unfairness*, highlighting the importance of imposing procedure fairness directly on the data generating process itself.
- We present the value instantiation rule that utilizes reference points in local causal modules to decouple the negative influence of objectionable data generating components, while keeping their neutral counterparts intact.
- Motivated by John Rawls's advocacy for *pure procedural justice* (Rawls, 1971; 2001), we configure reference points to the greatest benefit of the least advantaged individuals, so that the data generating process itself satisfies the requirements of procedural fairness.

## 2 PRELIMINARIES

In this section, we present our notation conventions and the requirements for procedural fairness.

### 2.1 NOTATIONS

We use uppercase letters to refer to variables, lowercase letters to refer to specific values that are taken by variables, bold-font letters to refer to list of variables, and calligraphic letters to refer to domains of values. For example, we denote all features by $\mathbf{Z}$, and denote a specific feature by $Z_i$ whose domain of value is $\mathcal{Z}_i$.[2] We denote the target variable by $Y$ and its predictor by $\widehat{Y}$.

For two random variables $W$ and $V$, we say that $W$ is a direct cause of $V$ if there is a change in distribution of $V$ when we apply an intervention on $W$ while holding all other variables fixed (Spirtes et al., 1993; Pearl, 2009). We can capture causal relations among variables via a directed acyclic graph (DAG) $\mathcal{G}$, where nodes (vertices) represent variables, and edges represent causal relations between variables and the corresponding direct causes. We use an ordered pair of nodes, e.g., $\rho = (W, V)$ to represent a directed edge $W \rightarrow V$. The atomic unit of objectionable and neutral components is an edge that symbolizes the causal relation between a node and its direct parent connected via that edge. One can apply the function $\mathrm{Tail}(\cdot)$ to get the tail node of an edge $\rho$, and apply $\mathrm{Parents}(\cdot)$ to list all direct parent nodes of a node in the graph.

### 2.2 REQUIREMENTS FOR PROCEDURAL FAIRNESS

There are different perceptions on the procedural emphasis of justice or fairness. *Perfect (imperfect) procedural justice* specifies a standalone criterion for the just outcome, and the existence (non-existence) of a feasible procedure guaranteed to lead to such just outcome (Rawls, 1971; 2001; Luce & Raiffa, 1989; Barry, 2010). The fair division is an example of *perfect procedural justice* (the person who divide the cake gets the last piece), and a trial is an example of *imperfect procedural justice* (the miscarriage of justice may occur not due to human faults, but as a result of an unforeseen

---

[2]Our framework naturally handles and benefits from more precisely defined disadvantaged individuals. The characterization of disadvantaged individuals may involve features including, but not limited to, the canonical protected features. For notation consistency with the literature, we reserve the letter $A$ for protected features.

confluence of circumstances that undermines legal purposes). *Pure procedural justice*, on the other hand, emphasizes that the procedure for determining the just result must actually be carried out, since there is no standalone criterion by which an outcome can be known to be just (Rawls, 1971; 2001).

Inspired by John Rawls's procedural conceptions of justice and the advocacy for *pure procedural justice* (Rawls, 1971; 2001), we address procedural fairness by incorporating requirements on data generating processes themselves. In order to avoid permitting too much in carrying out such procedure, additional requirements on the data generating process are imposed (Rawls, 1971; 2001):

Requirement I   *Fair Equality of Opportunity*
> The opportunity should be open and attainable, with the same prospects of success, for those who are at the same level of talent and ability, and have the same willingness to use them. Such equality of opportunity should not be influenced by arbitrary contingencies.

Requirement II   *The Difference Principle*
> The (social and economic) inequalities are to be arranged so that they are to the greatest benefit to the least advantaged members of the society.

The algorithmic fairness literature has presented proposals that more or less resonate with Requirement I and Requirement II. For example, Hardt et al. (2016b) propose *Equal Opportunity* to equalize true positive rates across groups, which is a group-level evaluation of the predicted outcome that shares the intuition behind Requirement I. *Minimax Group Fairness* (Martinez et al., 2020; Diana et al., 2021) measures the worst-case outcomes across groups, which aligns with the attention on state of affairs of disadvantaged individuals specified in Requirement II. Heidari et al. (2019a) propose a moral framework to evaluate fairness notions; the welfare analyses related to decision-making policies are conducted in both static and long-term settings (Heidari et al., 2018; 2019b).

Previous works articulate the intuition of procedural fairness primarily through emphases on outcomes or outcome-related statistics, rather than focusing on the procedural intricacies inherent to the data generating process itself. We would like to note that Requirement I and Requirement II are not intended to express standalone fairness criteria for the predicted outcome, and they serve as mandates for the data generating process to guarantee procedural fairness (Rawls, 1971; 2001). The violation of these requirements results in *disguised procedural unfairness*, and to the illustration we now turn.

## 3   ILLUSTRATING DISGUISED PROCEDURAL UNFAIRNESS

It has been widely recognized in the algorithmic fairness literature that causal analysis enables us to quantitatively audit and mitigate the discrimination in data generating process (Kusner et al., 2017; Kilbertus et al., 2017; Nabi & Shpitser, 2018; Chiappa, 2019; Wu et al., 2019; Creager et al., 2020; Nabi et al., 2022; von Kügelgen et al., 2022; Tang et al., 2023a). However, incorporating causal analysis alone does not automatically provide the immunity against *disguised procedural unfairness*. In this section, we reveal the commonly overlooked consequence of imposing fairness constraints directly on outcomes while narrowly addressing *only* objectionable aspects in data generation process.

For illustration, let us consider a linear model drawn from previous literature (Nabi & Shpitser, 2018; Chiappa, 2019), which involves variables $(A, C, M, L, Y)$ and is presented in Figure 1(a):

$$A \sim \text{Bernoulli}(p_A), \quad C = \epsilon_C, \quad M = \theta_A^M A + \theta_C^M C + \theta^M + \epsilon_M,$$
$$L = \theta_A^L A + \theta_C^L C + \theta_M^L M + \theta^L + \epsilon_L, \quad Y = \theta_A^Y A + \theta_C^Y C + \theta_M^Y M + \theta_L^Y L + \theta^Y + \epsilon_Y, \tag{1}$$

where $\epsilon_V \sim \mathcal{N}(0, \sigma_V^2), V \in \{C, M, L, Y\}$ are independent zero-mean Gaussian noise terms, and $\theta$'s are parameters in the linear model, with $\theta_W^V$ denoting the direct causal influence from $W$ to $V$.

After specifying the problematic paths in the data generating process, one can derive the path-specific effect (PSE) from the protected feature $A$ to the prediction $\widehat{Y}$ according to causal fairness notions (Kilbertus et al., 2017; Nabi & Shpitser, 2018; Chiappa, 2019; Wu et al., 2019; Nabi et al., 2019; 2022). Specifically, for objectionable components depicted by red edges in Figure 1(a), causal fairness notions impose requirements on the following quantity calculated based on model parameters:[3]

---

[3]In this linear example, the path-specific interventional causal effect and counterfactual causal effect share a same form in terms of the (sub-)structure of model parameter combinations, under the odds ratio scale (VanderWeele & Vansteelandt, 2010; Nabi & Shpitser, 2018) or the mean difference scale (Chiappa, 2019). Therefore, following Chiappa (2019), we use PSE without denoting the type of causal effect.

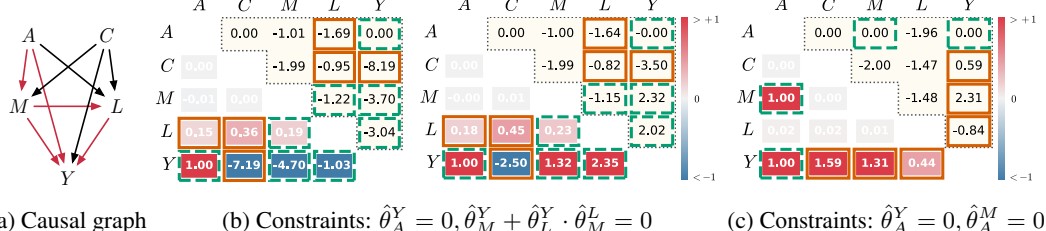

(a) Causal graph    (b) Constraints: $\hat{\theta}_A^Y = 0, \hat{\theta}_M^Y + \hat{\theta}_L^Y \cdot \hat{\theta}_M^L = 0$    (c) Constraints: $\hat{\theta}_A^Y = 0, \hat{\theta}_A^M = 0$

Figure 1: The linear example where causal fairness notions are applied. Panel (a) contains the causal graph for the data generating process. Panel (b) and panel (c) summarize the behavior of fitted parameters, where panel (b) corresponds to fairness constraints proposed by Kilbertus et al. (2017), and panel (c) corresponds to those proposed by Nabi & Shpitser (2018); Nabi et al. (2019; 2022). Orange solid-line boxes in the matrix are instantiations of the *disguised procedural unfairness* due to the violation of Requirement I.

$$\text{PSE} = \hat{\theta}_A^Y + \hat{\theta}_A^M (\hat{\theta}_M^Y + \hat{\theta}_L^Y \hat{\theta}_M^L). \tag{2}$$

Different causal fairness notions may employ different constraints to enforce fairness on the causal effect calculated in Equation (2). For instance, Kilbertus et al. (2017) consider the direct discrimination (the path $A \rightarrow Y$) and the proxy discrimination with respect to the unresolving variable $M$ (the paths $A \rightarrow M \rightarrow Y$ and $A \rightarrow M \rightarrow L \rightarrow Y$), and impose constraints $\hat{\theta}_A^Y = 0, \hat{\theta}_M^Y + \hat{\theta}_L^Y \cdot \hat{\theta}_M^L = 0$. Since the constrained optimization does not yield a unique solution apart from minor variations attributable to computational numerical errors, we present two sets of fitted parameters in Figure 1(b). Nabi & Shpitser (2018) propose to consider PSE as a whole and perform constrained optimization with PSE bounded by a small quantity (Nabi & Shpitser, 2018; Nabi et al., 2019; 2022). One sufficient condition is $\hat{\theta}_A^Y = \hat{\theta}_A^M = 0$, and the fitted parameters are presented in Figure 1(c).[4]

In the presented figures, the upper triangular entries (highlighted with a dotted contour) contain values of fitted parameters $\hat{\theta}$'s. The lower triangular entries contain signed relative deviations of the fitted parameter compared to the ground truth, i.e., $(\hat{\theta} - \theta)/|\theta|$, and therefore, the closer to 0 the value is, the more aligned with underlying process the fitted parameters are. The signed relative deviations in lower triangular entries are color-coded in a heat map format: light-gray indicates 0, shades of red indicate positive values, and shades of blue indicate negative values. The green dashed-line boxes denote objectionable components on which fairness constraints are enforced, and the orange solid-line boxes denote the unintentional deviations of $\hat{\theta}$'s from the ground truth $\theta$'s for neutral components.

As we can see from Figure 1(b) and Figure 1(c), the lower triangular matrices contain orange solid-line boxes, whose values significantly deviate from 0. It is tempting to think of such deviation as just an unintentional but inevitable consequence of solving the constrained optimization problem. However, if we identify certain components in the underlying data generating process as neutral, i.e., not objectionable, there is no guarantee that after introducing an arbitrary deviation (e.g., an increase or decrease in the linear coefficient), such component is still not objectionable.

Furthermore, even if one is relieved from the burden of providing justification behind the introduced deviation on neutral components, one would face yet another challenge of justifying the choice among different deviations. For example, the edge $C \rightarrow Y$ in Figure 1(a) represents a neutral data generating component. In each matrix in Figure 1(b) and Figure 1(c), if we refer to the second entry from left on the bottom row, we can see that the corresponding estimated parameter $\hat{\theta}_C^Y$ gets various amount of deviations. Among different values of the fitted parameter, there is no obvious reason why one should prefer any particular option over the others, even though they are derived by enforcing causal fairness notions that share the same underlying intuition, namely, to eliminate the discrimination along the paths $A \rightarrow Y$ and $A \rightarrow M \rightarrow \cdots \rightarrow Y$ in terms of PSE formulated in Equation (2) (Kilbertus et al., 2017; Nabi & Shpitser, 2018; Chiappa, 2019; Wu et al., 2019; Nabi et al., 2019; 2022).

---

[4]When applying *Path-Specific Counterfactual Fairness* or *PC-Fairness*, the derivation of $\widehat{Y}$ may involve additional counterfactual analyses instead of constraining model parameters, e.g., utilizing latent inference-projection (Chiappa, 2019) or deriving upper/lower bound for the counterfactual causal effect (Wu et al., 2019). These technical treatments are beyond the scope of the illustrative linear example.

Such arbitrary deviations from neutral components in the underlying process violate Requirement I, which prohibits the influence from arbitrary contingencies. As we shall see in Section 5.1, only enforcing fairness on objectionable components also invites the violation of Requirement II, since the inequality is not arranged to the greatest benefit of the least advantaged individuals.

## 4  DECOUPLING OBJECTIONABLE COMPONENTS FOR PROCEDURAL FAIRNESS

In Section 3, we demonstrate that causal fairness notions, which are among the most explicit procedural emphases on algorithmic fairness in current literature, do not offer the protection against *disguised procedural unfairness*. In this section, we present our framework of decoupling objectionable data generating components, and deriving the predicted outcome only with neutral components. We start in Section 4.1 by revisiting the linear example presented in Section 3. As an initial attempt to address *disguised procedural unfairness*, we consider a simple solution to avoid arbitrary alterations on neutral components. By reflecting on this simple approach, we aim to gain insights into the enforcement of procedural fairness requirements. Then, we present our approach in Section 4.2.

### 4.1  REFLECTING ON A SIMPLE APPROACH: AN INITIAL ATTEMPT

Let us revisit the linear example in Section 3. With a correct causal graph, a hypothesis class that is general enough, and a consistent estimator of model parameters, one can expect the estimated linear parameters to be as close as possible to the ground truth, if with an unlimited amount of data and without any fairness constraint imposed during optimization. One can then proceed by dropping estimated parameters that correspond to the objectionable components, i.e., setting to 0 the linear coefficients along red edges in Figure 1(a), and deriving the predicted outcome only with remaining parameters. Causal fairness notions are achieved by directly constraining PSE in Equation (2) to be zero (Kilbertus et al., 2017; Nabi & Shpitser, 2018; Nabi et al., 2019; Chiappa, 2019). Different from performing constrained optimization over model parameters (Section 3), this simple approach does not introduce arbitrary alterations on neutral components, and therefore, does not violate Requirement I. However, this approach should not be the general strategy to achieve procedural fairness, setting aside the consideration of Requirement II.

To begin with, the procedural conception of eliminating the influence of an edge in the causal graph does not always directly translate into constraining certain parameters in the model. If the model does not have additive structures among objectionable and neutral components, the decomposition in the format of dropping or modifying objectionable terms or parameters can not be easily carried out.[5] For complicated structures, e.g., a neural network, there is no principled way of pinpointing and separating parameters into disjoint sets that exclusively constitute objectionable or neutral components.

Furthermore, even if the model has additive structures among objectionable and neutral components, the fairness constraint may involve nonlinear relations among parameters, e.g., constraints proposed by Kilbertus et al. (2017) as presented in Figure 1(b). While dropping the term or setting the parameter to constants provides a sufficient condition to satisfy causal fairness notions, the solution may not be optimal. One may face further challenges of justifying the choice among possible solutions. Therefore, achieving procedural fairness calls for a more principled and scalable approach.

### 4.2  OUR FRAMEWORK

We consider the following question with respect to the causal mechanism: "What would have been the case if the objectionable component in the data generating process were ineffective, given that the model parameters (fitted without enforcing any fairness constraint) exhibit objectionable aspects?"[6] One way of tackling this problem is to isolate the parameter space that corresponds solely to objectionable components, such that fairness constraints can be applied only on objectionable components without affecting neutral ones. However, as we discussed in Section 4.1, this initial attempt does not appear to be a promising approach to achieve procedural fairness in general settings.

---

[5]Additive structures can take different forms, e.g., a linear combination of nonlinear but uncoupled terms (Hoyer et al., 2008), or a linear combination followed by nonlinear transformations (Zhang & Hyvärinen, 2009). Additive structures are not limited to models that only contain linear relationships.

[6]We provide additional discussions on the contrast between the counter-factual question with respect to causal mechanisms and that with respect to variables in Appendix B.1.

---

**Algorithm 1:** The Value Instantiation Rule for Local Causal Modules

---

**Input** : The $d_{\text{in}}(V; \mathcal{G})$-ary function $h_V\big(\text{Parents}(V); \hat{\theta}_V\big)$ modeling the causal mechanism between the node $V$ and its direct parents, where $d_{\text{in}}(V; \mathcal{G})$ is the the number of direct parents (in-degree) of $V$ in the graph $\mathcal{G}$. The configuration function $\text{ReferencePoint}(\cdot)$, which maps a directed edge corresponding to an objectionable component $\rho \in \mathcal{E}_{\text{Obj}}$ to a reference point (Definition 4.1) with the domain of value of the tail node of the edge.

**Output** : The derivation of the predicted outcome $\widehat{V}$ in the local causal module.

1 **If** *there is additional assumption on the functional form $\widetilde{h}_V(\cdot)$ and/or parameters $\widetilde{\theta}_V$* **Then**
2 $\quad$ $\hat{\theta}_V \leftarrow \widetilde{\theta}_V, h_V \leftarrow \widetilde{h}_V$ ; `// direct correction of the causal mechanism`
3 **Else**
4 $\quad$ **ForEach** *parent node $W_j$ in* $\text{Parents}(V) = (W_1, W_2, \ldots, W_{d_{\text{in}}(V; \mathcal{G})})$ **Do**
5 $\quad\quad$ **If** *the edge $\rho_j = (W_j, V) \in \mathcal{E}_{\text{Obj}}$, i.e., $W_j \rightarrow V$ is an objectionable component* **Then**
6 $\quad\quad\quad$ $w_j$ gets the value $\text{ReferencePoint}(\rho_j)$, because $W_j = \text{Tail}(\rho_j)$;
7 $\quad\quad$ **Else If** *there is at least one ancestor nodes of $W_j$ was set to a reference point* **Then**
8 $\quad\quad\quad$ $w_j$ gets the value that $W_j$ would have taken as a downstream of its ancestor nodes, to which reference points, if any, have been assigned;
9 $\quad\quad$ **Else**
10 $\quad\quad\quad$ $w_j$ gets the value of variable $W_j$ for the record in the data set;
11 $\widehat{v} \leftarrow h_V(w_1, w_2, \ldots, w_{d_{\text{in}}(V; \mathcal{G})}; \hat{\theta}_V)$.

---

Ideally, if there is additional knowledge or assumption about how an objectionable component can be corrected in terms of the functional form of the causal mechanism, one can directly replace that objectionable component in the model with its neutral version to derive the fair prediction. In practical scenarios, such information may not be readily available, which makes the direct correction of objectionable causal mechanisms not always a viable option. We need to find an alternative way to decouple objectionable components from the data generating process.

Instead of focusing on model parameters and trying to isolate objectionable components during parameter fitting, we turn our attention to the inputs of objectionable components. We explore the possibility of finding appropriate input values for local causal modules, without enforcing any fairness constraint at the stage of learning model parameters. We recognize and remain aware that certain components of the data generating process are objectionable, and our framework consists of two parts: the specification of the value instantiation rule to properly propagate the value of upstream variables to the downstream (Section 4.2.1, fulfilling Requirement I), and the configuration of reference points for input nodes that correspond only to objectionable components, in accordance with the greatest benefits of the least advantaged individuals (Section 4.2.2, fulfilling Requirement II).

### 4.2.1 THE VALUE INSTANTIATION RULE FOR LOCAL CAUSAL MODULES

Local causal modules, which depict the causal relation between a node and its direct parent nodes, are irrelevant to each other because of the causal modularity in static settings.[7] More specifically, the manipulation in one local causal module, e.g., the modification of the functional form in the corresponding causal relation, will not affect the causal mechanism in other modules.

**Definition 4.1 (Reference Point).** A reference point is a fixed value that a node propagates along an edge. A node is set to a reference point if and only if it is the tail node of an objectionable component. When the node is the tail for multiple objectionable components, it may be set to different reference points, each of which corresponds to one objectionable component within its local causal module.

We regard each node as a placeholder and assign appropriate values to the input according to the value instantiation rule, which contains a sequence of options as summarized in Algorithm 1. Depending on the applicability of each option in the presented order (Steps 5 – 10 of Algorithm 1), each input

---

[7]Causal modularity, also known as exogeneity (Engle et al., 1983) or independence of causal mechanism (Peters et al., 2017), is the direct outcome of the causal Markov condition for the corresponding DAG in static settings (Spirtes et al., 1993; Pearl, 2009). The cases involving dynamic settings or direct cyclic graphs (DCGs) are beyond the scope of the current work.

---

**Algorithm 2:** Aggregating Local Causal Modules while Decoupling Objectionable Components

---

**Input** : The data set $\mathcal{D}$, the hypothesis class $\mathcal{H}$ and the parameter space $\Theta$, the causal graph $\mathcal{G} = (\mathbf{V}, \mathcal{E})$, the list of index $\mathcal{I}$ for all nodes $\mathbf{V}$. The set of edges $\mathcal{E}_{\mathrm{Obj}}$ where each edge corresponds to an objectionable component. The $\mathrm{ReferencePoint}(\cdot)$ configuration.

**Output** : The derivation of the predicted outcome $\widehat{Y}$ that *decouples* objectionable components from the data generating process, and *only* makes use of neutral components.

1 Sort the list of index $\mathcal{I}$ such that parent nodes, if any, appear before the node itself;

2 **ForEach** *node index $i \in \mathcal{I}$* **Do**                          // learn model parameters

3    **If** *the number of direct parents of node $V_i$, i.e., the in-degree, $d_{\mathrm{in}}(V_i; \mathcal{G}) > 0$* **Then**

4       Fit model parameters in the local causal module between $V_i$ and its direct parent nodes $\mathrm{Parents}(V_i)$, without any fairness constraint:

$$h_{V_i}, \hat{\theta}_{V_i} \leftarrow \operatorname*{argmin}_{\theta \in \Theta, h \in \mathcal{H}} \mathcal{L}_{V_i}\big(h(\mathrm{Parents}(V_i); \theta), V_i\,; \mathcal{D}\big), \mathcal{L}_{V_i} \text{ is the loss function for } V_i;$$

5 According to the sorted list of node index $\mathcal{I}$, apply the value instantiation rule (Algorithm 1) to each local causal module in sequence, and then derive prediction $\widehat{Y}$ according to Equation (3).

---

node can be set to a reference point, or the value attained as the downstream of reference point(s), or by default the original value in the data set when none of the prior options apply.[8]

The value instantiation rule can effectively and correctly decouple the influence of objectionable components. To begin with, a clear boundary exists between the input for an objectionable component and that for a neutral component. In a local causal module, the scope of consideration is limited to the causal relation between the output node and its direct parents. The causal relation is in turn represented by edges that share the output variable as the head node, and inputs as tail nodes. Because there is only one output node in the local causal module and there can be at most one edge between any pair of nodes in the graph, each input node can only be the tail of one edge, which represents either an objectionable component or a neutral one but not both. This distinction enables the value instantiation rule to address objectionable components while keeping neutral ones intact.

Besides, the mapping from input to output for each local causal module is shaped not only by the fitted model parameters, but also by the applicable value instantiation option for each input node. Propagating value along certain edges has been characterized in the causal inference literature (Robins & Greenland, 1992; Pearl, 2001; Shpitser & Tchetgen, 2016; Peters et al., 2017). Particularly, Shpitser & Tchetgen (2016) present a graphical hierarchy of causal interventions and formally introduce terms "edge intervention" and "path intervention", where the intervention is carried out along certain edges or paths, as natural refinements of the canonical definition of causal intervention, i.e., the "node intervention". From a purely technical point of view, when there is no direct correction of the causal mechanism, the value instantiation rule is the edge-specific version of causal intervention. We set reference points for tail nodes of certain edges in order to change the overall input-output relation of local causal modules. Our utilization of value instantiation rule aims to decouple objectionable data generating components and satisfy procedural fairness requirements, and this is very different from performing causal inference to quantify causal effects among certain variables.

### 4.2.2 THE CONFIGURATION OF REFERENCE POINT VALUES

In this subsection, we first present how to utilize Algorithm 1 in each local causal module and derive the final prediction, when the mapping $\mathrm{ReferencePoint}(\cdot)$ is available and the entire causal graph is taken into consideration. Then, we present how to obtain the mapping $\mathrm{ReferencePoint}(\cdot)$, i.e., the configuration of reference point values in accordance with Requirement II for procedural fairness.

Because of the modularity property, the value propagations in different local causal modules do not interfere with each other and are fully specified in Algorithm 1. Since there is no loop in DAGs, we sort the nodes in the graph such that parent nodes appear before the node itself. If we apply the value instantiation rule to each local causal module in this order, we can correctly keep track of reference points as well as their downstream effects across the entire data generating process. We summarize in Algorithm 2 the procedure of aggregating local causal modules while decoupling objectionable data

---

[8]We present detailed illustrations of the application of value instantiation rule in Appendix C.

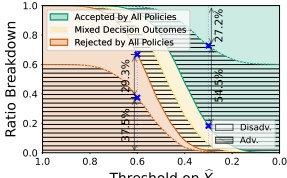 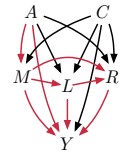 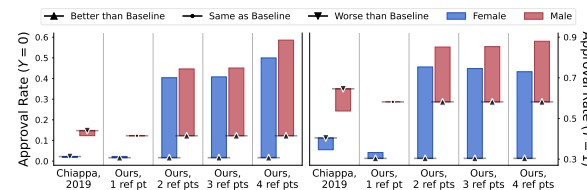

(a) Revisit linear example     (b) UCI Adult     (c) Summary of results on the UCI Adult data set

Figure 2: Experimental results on the simulated data and the real-world UCI Adult data set. Panel (a) demonstrates the *disguised procedural unfairness* due to the violation of Requirement II. Panel (b) presents the causal graph for UCI Adult data set. Panel (c) summarizes results on UCI Adult data set, where we present comparisons between group-wise approval rates for low-/high- income individuals, before and after fairness considerations are implemented.

components. One can derive the predicted outcome $\widehat{Y}$ for an individual with features $\mathbf{Z} = \mathbf{z}$:

$$\widehat{y} = \underset{i \in \mathcal{I}}{\circ} \big(h_{V_i} \circ \text{ReferencePoint}\big)(\mathbf{z}; \mathcal{E}_{\text{Obj}}, \hat{\theta}_{V_1}, \ldots, \hat{\theta}_{V_{|\mathcal{I}|}}), \tag{3}$$

where $h_{V_i} \circ \text{ReferencePoint}$ is the composite function denoting the input-output relation of the local causal module after applying the value instantiation rule (Algorithm 1), and the function composition operation $\circ(\cdot)$ is performed over the sorted list of node indices $i \in \mathcal{I}$.

Deriving predicted outcome while decoupling objectionable data generating components involves a composite function of fitted local causal modules and the $\text{ReferencePoint}(\cdot)$ configurations. In other words, the exact value of predicted outcome after introducing reference points cannot be predetermined before specifying individual's features and actually carrying out the derivation procedure outlined in Algorithm 2. To obtain the mapping $\text{ReferencePoint}(\cdot)$ in accordance with Requirement II of procedural fairness, we focus on the specified least advantaged individuals. We configure the value of reference points such that they can yield the most beneficial predicted outcome for least advantaged individuals (assuming the larger the $\widehat{Y}$, the more favorable the outcome):

$$\text{ReferencePoint} = \underset{f:\mathcal{E}_{\text{Obj}} \to \mathbf{\Omega}}{\text{argmax}} \underset{\text{least advantaged individuals}}{\mathbb{E}} \big[\widehat{Y}\big],$$
$$\text{s.t.} \quad \mathbf{\Omega} = \underset{\rho \in \mathcal{E}_{\text{Obj}}}{\times} \big((\Omega \circ \text{Tail})(\rho)\big), \quad \text{and} \quad \widehat{Y} = \underset{i \in \mathcal{I}}{\circ} \big(h_{V_i} \circ f\big)(\mathbf{Z}; \mathcal{E}_{\text{Obj}}, \hat{\theta}_{V_1}, \ldots, \hat{\theta}_{V_{|\mathcal{I}|}}), \tag{4}$$

where $\Omega(\cdot)$ denotes the domain of value of a node. Given an edge in the set of objectionable components $\rho \in \mathcal{E}_{\text{Obj}}$, $\text{ReferencePoint}(\rho)$ ranges over $(\Omega \circ \text{Tail})(\rho)$, the domain of value of the tail node of edge $\rho$. Since there is a finite number of edges in $\mathcal{E}_{\text{Obj}}$, the codomain of $\text{ReferencePoint}(\cdot)$, denoted as $\mathbf{\Omega}$, can be expressed as the Cartesian product of $(\Omega \circ \text{Tail})(\rho)$ for all edges $\rho \in \mathcal{E}_{\text{Obj}}$. Therefore, given $f$ in the space of functions for $\text{ReferencePoint}(\cdot)$, the predicted outcome $\widehat{Y}$ is obtained as specified in Equation (4), whose form is similar to Equation (3) but with the function $\text{ReferencePoint}(\cdot)$ replaced by $f(\cdot)$ during optimization.

## 5 EXPERIMENTS

In this section, we present experimental results on both simulated and real-world data. In Section 5.1, we demonstrate how causal fairness notions can violate Requirement II of procedural fairness. In Section 5.2, we present experimental results on UCI Adult data set (Becker & Kohavi, 1996).[9]

### 5.1 ILLUSTRATING REQUIREMENT II VIOLATION: REVISITING LINEAR EXAMPLE

In Figure 2(a), we revisit the illustrative linear example presented in Section 3 and quantitatively demonstrate the violation of Requirement II. We summarize the predicted outcomes by different

---

[9]Due to space limit, we present in Appendix D implementation details, the discussion on scalability, further experimental details and additional results on simulated and real-world data sets, including the UCI Adult (Becker & Kohavi, 1996) and the Folktables (Ding et al., 2021) data sets. Our implementation can be found in the Github code repository: `https://github.com/zeyutang/DecoupleObjectionable`.

decision-making policies, where model parameters are optimized with or without fairness constraints. We include constraints presented in Figure 1(b) and Figure 1(c), and consider whether there exist individuals who are always rejected no matter which fairness notion is enforced. To derive a binary decision, we vary the threshold from $1.0$ to $0.0$ to reflect the abundance of resource (from scarce to plentiful). As we can see from Figure 2(a), although the overall approval rate increases as the resource becomes more ample, the disadvantaged group proportionally suffers more among those who are rejected by all decision policies, and at the same time, prospers less among those who are accepted by all policies, barring marginal representation imbalance. This indicates the violation of Requirement II in addition to Requirement I (presented in Section 3), since the inequality is not arranged to the benefit of the least advantaged individuals.

## 5.2 Results on UCI Adult Data Set

In Figures 2(b) and 2(c), we present the causal modeling and the summary of results on UCI Adult data set (Becker & Kohavi, 1996). Following Nabi & Shpitser (2018) and Chiappa (2019), the paths $A \to Y$ (from `sex` to `income`) and $A \to M \to \cdots \to Y$ (from `sex`, mediated by `marital status`, to `income`) are problematic paths. We present experimental results when applying different approaches. Specifically, Chiappa (2019) first fits model parameters without any fairness constraints, and then incorporates additional latent variables and parameters in variational reasoning, aiming to reconstruct descendant variables of $A$ along problematic pathways with the latent inference-projection approach. Our framework treats the red edges in Figure 2(b) as potential locations for objectionable components, and consider reference point configurations of different strengths, according to the number of decoupled objectionable components.

As we can see from Figure 2(c), compared to the unconstrained baseline, the state of affairs are downgraded in terms of the approval rate when applying *Path-Specific Counterfactual Fairness* (Chiappa, 2019), where `sex` is flipped to construct the counterfactual. In the left chart of Figure 2(c), i.e., when $Y = 0$, the very low baseline approval rate of the least advantaged individuals (here, is the female group) is further decreased. In contrast, our framework utilizes the neutral components in the data generating process, and makes use of the reference points derived to the benefit of the least advantaged individuals. Compared to the unconstrained optimized baseline, our results provide more opportunities for the least advantaged individuals, offering boosts of approval rates by decoupling the objectionable components in the data generating process.

Our results also indicate the potential limitation of only focusing on protected features as in previous literature. We propose to consider all objectionable components, which include, but not limited to, those that take the protected feature as input. For instance, we observe that when objectionable components consist of edges $A \to Y$ (from `sex` to `income`) and $M \to Y$ (from `marital status` to `income`), the reference points actually do not flip `sex` $A$ from female to male, and only specify `marital status` as "married". This indicates that in the presence of objectionable data generating components, the discrimination is not mitigated by treating female individuals as males, but by perceiving (from decision-maker's perspective) all individuals (male and female) as if they were females along $A \to Y$, and married along $M \to Y$.

## 6 Concluding Remarks

In this paper, we focus on procedural fairness in terms of the requirements on the data generating process of the prediction model. We reveal and address the frequently overlooked issue of *disguised procedural unfairness*. In particular, previous approaches can introduce arbitrary alterations on neutral components of data generating process (violating Requirement I); the predicted outcome can exhibit inequalities that are not to the greatest benefit of the least advantaged individuals (violating Requirement II). In accordance with the requirements of procedural fairness, we propose a framework that utilizes reference points together with appropriate value instantiation rule.

We highlight the importance and necessity of decoupling objectionable data generating components to achieve procedural fairness.[10] Future works naturally include developing efficient and effective strategies for decoupling objectionable components if additional knowledge or assumption about the underlying data generating process can be utilized in various practical scenarios.

---

[10]We provide further discussions on implications and potential limitations of our approach in Appendix E.

ETHICS STATEMENT

The motivation of our work is to pursue procedural fairness with respect to the data generating process. The research is performed under full awareness of, and with adherence to, ICLR Code of Ethics. We reveal and address the frequently overlooked issue of *disguised procedural unfairness*. Our framework decouples objectionable components from the data generating process, aiming to ensure the fulfillment of requirements for procedure fairness. We hope our work can draw attention to the important issue of *disguised procedural unfairness*, and inspire further research to promote procedural guarantees on fairness of the data generating process for prediction or decision-making.

REPRODUCIBILITY STATEMENT

We provide the implementation of our framework, along with the instruction on how to run it, in the Github code repository (also provided in Section 5): `https://github.com/zeyutang/De coupleObjectionable`.

ACKNOWLEDGEMENT

This material is based upon work supported by the AI Research Institutes Program funded by the National Science Foundation (NSF) under AI Institute for Societal Decision Making (AI-SDM), Award No. 2229881. This project is also partially supported by an Amazon Research Award (Fall 2022 CFP), the National Institutes of Health (NIH) under Contract R01HL159805, and grants from Apple Inc., KDDI Research Inc., Quris AI, and Infinite Brain Technology.

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

# SUPPLEMENT TO "PROCEDURAL FAIRNESS THROUGH DECOUPLING OBJECTIONABLE DATA GENERATING COMPONENTS"

**Zeyu Tang**[1], **Jialu Wang**[2], **Yang Liu**[2,4], **Peter Spirtes**[1], and **Kun Zhang**[1,3]

[1]Department of Philosophy, Carnegie Mellon University
[2]Computer Science and Engineering Department, University of California, Santa Cruz
[3]Machine Learning Department, Mohamed bin Zayed University of Artificial Intelligence
[4]ByteDance Research
`zeyutang@cmu.edu`, {`faldict, yangliu`}`@ucsc.edu`, {`ps7z@andrew., kunz1@`}`cmu.edu`

## TABLE OF CONTENTS: APPENDIX

Table 1: Summary of comparisons between our approach and closely related previous works.

| Fairness considerations | Address *disguised procedural unfairness* | Not depend on causal effect identifiability | Individualized mitigation or actionable recourse | Individual-level evaluation and auditing | Precisely defined disadvantaged individuals |
|---|---|---|---|---|---|
| (Conditional) independence relationships (Dwork et al., 2012; Hardt et al., 2016b; Zafar et al., 2017; Coston et al., 2020; Imai & Jiang, 2020; Mishler et al., 2021) | ✗ | depending on the definition | ✗ | ✗ | ✗ |
| Path-specific (interventional) causal effect (Kilbertus et al., 2017; Zhang et al., 2017; Nabi & Shpitser, 2018; Nabi et al., 2019; 2022; Salimi et al., 2019) | ✗ | ✗ | ✗ | ✓ | ✗ |
| (Path-specific) counterfactual causal effect (Kusner et al., 2017; Chiappa, 2019; Wu et al., 2019) | ✗ | ✗ | ✗ | ✓ | ✗ |
| Cost/effort incurred on users to perform recourse (Ustun et al., 2019; Gupta et al., 2019; von Kügelgen et al., 2022) | ✗ | ✗ | ✓ | ✓ | ✗ |
| Intersectional definition of subgroups (Kearns et al., 2018; Foulds et al., 2020) | ✗ | ✓ | ✗ | ✗ | ✓ |
| **Our approach** | ✓ | ✓ | ✓ | ✓ | ✓ |

## A  Previous Works on Algorithmic Fairness Related to Data Generating Process

In this section, we review related works in the previous literature that draw connections between algorithmic fairness and data generating process, including causal fairness notions (Section A.1), responsive agents (Section A.2), and representation learning with fairness considerations (Section A.3). We summarize the comparisons between our approach and the previous literature in Table 1. Detailed discussions of the differences and connections are presented in Section B.

### A.1  Causal Fairness Notions

The algorithmic fairness literature has recognized that causal analysis provides a principled way to quantify the discrimination in data generating process. There are different proposals with respect to the object of interest when evaluating causal fairness.

**Path-Specific Interventional Causal Effects**  Motivated by the idea of quantifying the causal influence from the protected feature $A$ to final outcome $Y$ according to the type of mediating variables (if any), Kilbertus et al. (2017) propose to consider disjoint sets of descendant variables of $A$. Particularly, there are descendant variables that are influenced by the protected feature $A$ in an unproblematic way, and these variables are called "resolving variables"; there are also descendant variables that pass on influence from $A$ to $Y$ in an unjustifiable manner, and these variables are called "proxies". Kilbertus et al. (2017) capture fairness through the nonexistence of paths from $A$ to $Y$ that are not blocked by a "resolving" mediator (*No Unresolved Discrimination*). Nabi & Shpitser (2018) and follow-up works (Nabi et al., 2019; 2022) propose to directly quantify path-specific interventional causal effect from the protected feature $A$ to the final outcome $Y$ along problematic paths, and define causal fairness in terms of the nonexistence or bounded value of such path-specific causal effect.

**(Path-Specific) Counterfactual Causal Effects**  Apart from the interventional causal effect, there are proposals that consider the contrast between current world and hypothetical world, and define causal fairness accordingly. Kusner et al. (2017) consider the counterfactual value that the protected feature $A$ can take, and reason about the difference between the prediction/decision made in the

current world and that made in the counterfactual world. The data generating process for the predicted value or the decision outcome satisfies *Counterfactual Fairness* if the aforementioned difference is eliminated, taking into consideration all paths in the causal graph that start from the protected feature $A$ and end with the final outcome $Y$ (Kusner et al., 2017). Chiappa (2019) and Wu et al. (2019) concurrently present the more fine-grained version of *Counterfactual Fairness*, namely, *Path-Specific Counterfactual Fairness* or *PC-Fairness*, and propose that the evaluation of counterfactual causal effect from $A$ and $Y$ can be carried out in a path-specific way.

**Causal Quantities and Observed Disparities**   Different from evaluating causal fairness violation in terms of the causal effect from attributes (e.g., the protected feature $A$) to the final outcome $Y$, there are also proposals that put more emphasis on disparities in error rates related to potential outcomes (Rubin, 1974; 2005). Zhang & Bareinboim (2018b) present a decomposition of the observed disparity in terms of counterfactual quantities, and show that one can utilize the *causal explanation formula* to decompose the total variation between the protected feature $A$ and the final outcome $Y$ in terms of counterfactual direct effect, counterfactual indirect effect, and counterfactual spurious effect (Zhang & Bareinboim, 2018b). Zhang & Bareinboim (2018a) focus on the group-level fairness notion *Equalized Odds* (Hardt et al., 2016b) and present the decomposition of group-level true/false positive rate disparities in terms of counterfactual direct error rates, counterfactual indirect error rates, and counterfactual spurious error rates (Zhang & Bareinboim, 2018a). Previous literature also contains causal analogues of group-level observational disparity measures, which capture conditional independence relationships among the protected feature, the final outcome, and the potential outcome, for instance, *Counterfactual Predictive Parity* (Coston et al., 2020), *Counterfactual Equalized Odds* (Mishler et al., 2021), and *Conditional Principal Fairness* (Imai & Jiang, 2020).

## A.2    SCENARIOS WITH RESPONSIVE AGENTS

When deploying a decision-making policy, the subject of the decision might respond to the predicted outcome or decision. Previous literature has formulated such responsive behavior of agents as well as the potential fairness implications from the perspective of distribution shifts (Coston et al., 2019; Singh et al., 2021; Rezaei et al., 2021; Chen et al., 2022; 2023), the strategic behavior of individuals (Hardt et al., 2016a; Hu et al., 2019; Milli et al., 2019; Perdomo et al., 2020; Estornell et al., 2023), and the actionable recourse or personal efforts of individuals (Ustun et al., 2019; Gupta et al., 2019; Heidari et al., 2019b; von Kügelgen et al., 2022). Compared to causal fairness notions where the goal is to quantify the causal influence from attributes to the final outcome, the analysis with respect to responsive agents or population have different emphases when considering the role of data generating process. For instance, one can explicitly model the underlying data generating process and compare costs of actions an individual can possibly take, and then evaluate fairness in terms of the disparity of efforts incurred on the individual in order to obtain favorable decisions (von Kügelgen et al., 2022). This is different from constructing counterfactuals and reason about the causal effect of interest from certain attributes to the final outcome (Section A.1).

## A.3    FAIR REPRESENTATION LEARNING

The algorithmic fairness literature also includes fairness considerations in the context of representation learning. Zemel et al. (2013) aim at deriving the fair representation that encodes the data as good as possible, and at the same time obfuscates the group membership of the agents. Louizos et al. (2016) focus on the variational autoencoding (VAE) architecture (Kingma & Welling, 2014) and propose to utilize VAEs with priors that encourage independence between the protected feature and latent factors of variation, so that downstream tasks can be performed on "purged" latent representations. Madras et al. (2018) consider the setting where the learned representation is utilized by downstream tasks with unknown objectives, and explore the connection between observational group fairness notions and adversarial representation learning. Locatello et al. (2019) consider the setting where the prediction is based on the learned representation of (potentially high-dimensional) observations and the protected feature is unobserved. Locatello et al. (2019) investigate the effectiveness of different proposals of disentanglement on various state-of-the-art models, and suggest the potential of encouraging certain fairness notions through disentangled representation learning when the protected feature in not observed. Various technical treatments of the fairness violation are also proposed, for instance, the information-theoretically motivated objective for learning controllable (in the sense

of expressiveness-fairness tradeoff) fair representations (Song et al., 2019), the lower bounds on group-wise or joint errors of any (approximately) fair classifier (Zhao & Gordon, 2022), the balanced error rates and conditional alignment of representations (Zhao et al., 2020), the bi-level optimization with implicit path alignment (Shui et al., 2022).

# B  OUR APPROACH: DIFFERENCES AND CONNECTIONS TO PREVIOUS WORKS

In this section, we present differences and connections of our approach compared to previous works. In particular, we discuss the contrast between counter-factual analysis with respect to the variable and that with respect to the causal mechanism (Section B.1), the difference between the role played by reference points in our framework and that played by agent's responses in previous works (Section B.2), the comparison between the holistic approach of fair representation learning with outcome emphasis on fairness, and the modular approach of decoupling objectionable data generating components with procedural emphasis on fairness (Section B.3).

## B.1  SUBJECT OF COUNTER-FACTUAL ANALYSIS: VARIABLE VS. LOCAL CAUSAL MECHANISM

Sharing the procedural emphasis on algorithmic fairness, our approach is most closely related to causal fairness notions proposed in the previous literature (Section A.1). To differentiate from the technical term "counterfactual" in the causal inference literature (Spirtes et al., 1993; Pearl, 2009; Peters et al., 2017), we use the term "counter-factual" with a hyphen, as the opposite to "factual", to express in a general sense that something has not in fact happened in the current world. The technical treatment may involve interventional (Kilbertus et al., 2017; Nabi & Shpitser, 2018; Nabi et al., 2019; 2022) and/or counterfactual (Kusner et al., 2017; Chiappa, 2019; Wu et al., 2019) causal effects.

**Question B.1 (Counter-Factual Analysis w.r.t. Variables Only).** Under certain conditions and assumptions, what would happen to the predicted outcome in the factual world and the counter-factual world, had certain **variables** taken different values?

At a high level, the primary question asked by previous causal fairness proposals is Question B.1. As we have seen in Section A.1, causal fairness notions are based on estimating or bounding certain causal effects among variables. Typically, the variables involved in the causal effect of interest include the protected feature $A$, the final outcome $Y$ or its predicted counterpart $\widehat{Y}$, and certain specific variables closely related to $A$ but not the protected feature itself, e.g., explanatory features (Kamiran et al., 2013), proxy variables (Kilbertus et al., 2017), redlining attributes (Zhang et al., 2017), admissible variables (Salimi et al., 2019), and so on. Apart from technical details of the quantification, causal fairness notions propose that the (combination of) causal effects from the protected feature or proxy variables to the outcome should be bounded around a certain constant.

Although the quantification of the causal effect relies on certain assumption or knowledge of the underlying data generating process (e.g., the causal modeling in terms of a DAG), the fairness violation is directly quantified in terms of causal effects on the outcome $Y$ or $\widehat{Y}$. Such quantification involves introducing counter-factual values for the variables, specified in an *ex ante* way. For instance, it is a common starting point for causal fairness notions to consider the domain of values for the protected feature $A$ (Kilbertus et al., 2017; Kusner et al., 2017; Nabi & Shpitser, 2018; Nabi et al., 2019; 2022; Chiappa, 2019; Wu et al., 2019), and then define the causal effect on outcome accordingly.

While previous causal fairness notions are proposed with procedural emphasis on algorithmic fairness, they are carried out through counter-factual analyses with respect to variables instead of the data generating process itself, and therefore, resonate more with the outcome emphasis on fairness. We argue that *procedural fairness* necessitate counter-factual analyses with respect to causal mechanisms instead of only variables. We consider the following question:

**Question B.2 (Counter-Factual Analysis w.r.t. Local Causal Mechanisms).** Under certain conditions and assumptions, what would happen to the predicted outcome in the factual world and the counter-factual world, had certain **local causal mechanisms** behaved differently?

Te begin with, our work is not to propose a causal fairness notion defined directly on the outcome (counter-factual analysis with respect to variables), but a framework to decouple objectionable

data generating components for procedural fairness (counter-factual analysis with respect to causal mechanisms). Although the causal modeling of the data generating process is explicitly considered by causal fairness notions, previous works focus on Question B.1 and have not adequately addressed the procedural guarantee of fairness. As a consequence, the issue of *disguised procedural unfairness* is often overlooked (illustrated in Section 3), undermining the intention to achieve procedural fairness. In contrast, our framework does not directly enforce fairness on the outcome. We consider Question B.2 and utilize the value instantiation rule for local causal modules (Algorithm 1) together with appropriate reference points in the overall pipeline (Algorithm 2), to ensure that the requirements of procedural fairness are satisfied.

Furthermore, our framework is not a special case nor an extension to previous causal fairness notions. We reflect on the goal of achieving procedural fairness and propose to decouple objectionable data generating components from their neutral counterparts when performing prediction. We first specify the value instantiation rule for local causal modules, and then find appropriate values for reference points in an *ex post* way, for the purpose of decoupling the corresponding objectionable data generating components. This is very different from intervening on the input variable and setting values in an *ex ante* way, for the purpose of deriving interventional or counterfactual causal effect on downstream variables.

Moreover, our framework can handle more precise definitions of the disadvantaged individuals. In practical scenarios, the least advantaged individuals are not always precisely and fully characterized by the value of the protected features (Crenshaw, 1990; Kearns et al., 2018; Foulds et al., 2020). Because previous causal fairness notions focus on Question B.1, more precise definitions of the disadvantaged individuals introduce additional technical difficulties, especially the identifiability of causal effects. In contrast, we consider Question B.2, and our framework naturally handles, and in fact, benefits from the more precise specifications of the least advantaged individuals. As we shall see in Section D, we can focus on the state of affairs of the least advantaged individuals and further derive the reference points to their greatest benefit by solving the optimization problem in Equation (4).

## B.2 AGENT'S RESPONSE VS. ASSIGNED REFERENCE POINT

In terms of the possible discrepancy between the actual value and the perceived value (e.g., as seen by a decision-making policy) of individual's attributes, the reference points in our framework share a similar flavor with the updated attribute values discussed in previous works on responsive agents (Section A.2). Different from the consideration of the fairness implication of strategic behaviors on the decision-making policy (Hu et al., 2019; Milli et al., 2019; Estornell et al., 2023) or the impartiality of the effort for an actionable recourse (Ustun et al., 2019; Gupta et al., 2019; Heidari et al., 2019b; von Kügelgen et al., 2022), we do not consider potential responses (based on benevolent or malicious intentions) to the deployed or to-be-deployed decision-making policy. Instead, we focus on the data generating process itself, and investigate how objectionable components can be decoupled from the process to achieve procedural fairness.

For objectionable data generating components, when there is no additional assumption on the functional form of the fair local causal module, reference points are introduced to counteract and decouple the objectionable components. From the decision-maker's perspective, when it is acknowledged that objectionable data generating components exist, reference points instruct how the decision-making policy should use as inputs for these components (instead of the actual data). Such perception of attribute values is for the purpose of changing the behavior of the local causal module that contains problematic aspects, so that objectionable data generating components can be decoupled.

## B.3 HOLISTIC FAIR REPRESENTATION LEARNING VS. MODULAR OBJECTIONABLE COMPONENTS DECOUPLING

The high-level goal of fair representation learning is to derive an encoding to express the data, as a replacement for directly making use of the observational features (Section A.3). The formulation of the problem aims to provide readily available representations that can be utilized by third-parties for downstream tasks. The representation is derived with a holistic approach, treating the entire feature map as a whole when enforcing certain fairness notions.

Because of the intended obliviousness, or robustness to a certain extent, of knowledge or assumption about downstream tasks that utilize the learned representations, the criterion of interest for fair representation learning is largely limited to associative (i.e., not causal) fairness notions. Specifically, *Demographic Parity* (Calders et al., 2009; Dwork et al., 2012) in introduced to limit the marginal dependence between the (potentially unobserved) protected feature and the final outcome (Zemel et al., 2013; Louizos et al., 2016; Madras et al., 2018; Locatello et al., 2019; Song et al., 2019; Zhao & Gordon, 2022). *Equalized Odds* (Hardt et al., 2016b) is considered to enforce the (approximate) conditional independence between the protected feature and the predicted outcome given the ground truth (Zhao et al., 2020). *Predictive Parity* (Dieterich et al., 2016; Chouldechova, 2017; Zafar et al., 2017) is utilized to promote the sufficiency condition, i.e., the (approximate) conditional independence between the protected feature and the ground truth given the predicted outcome (Shui et al., 2022).

The similarity between our approach and previous works on fair representation learning lies in the fact that there is no fairness-related constraint on the predictor itself. The reasons behind this similarity are multifaceted. Fair representation learning aims to provide downstream-task-ready feature maps, often without an explicit reference to properties of the predictor or decision-making policy. In contrast, our framework specifically addresses the *disguised procedural unfairness*. The requirements of procedural fairness prohibit arbitrary alterations on neutral data generating components, and necessitate decoupling objectionable components with appropriate reference points. Furthermore, different from the holistic approach of deriving representation with outcome emphasis on fairness, we aim to provide the procedural fairness guarantee. We take a modular approach and investigate objectionable aspects of the data generating process in each local causal mechanism.

## C    DETAILED ILLUSTRATIONS OF OUR FRAMEWORK

In this section, we present detailed illustrations of the value instantiation rule (Algorithm 1) and the overall pipeline of decoupling objectionable components (Algorithm 2). We first present the step-by-step application of our framework in Section C.1. Then, in Section C.2, we provide remarks on derivation details.

### C.1    A WORKED-OUT EXAMPLE

Let us consider the following data generating process involving variables $(A, X_1, X_2, X_3, X_4, Y)$:

$$
\begin{aligned}
A &= E_A, \\
C &= E_C, \\
X_1 &= f_{X_1}(A, C, E_1), \\
X_2 &= f_{X_2}(A, C, X_1, E_2), \\
X_3 &= f_{X_3}(C, X_2, E_3), \\
X_4 &= f_{X_4}(X_1, X_2, E_4), \\
Y &= f_Y(A, X_1, X_2, X_3, X_4, E_Y),
\end{aligned}
\tag{5}
$$

where $E.$'s are independent noise terms, and $f.$'s specify the functional form of causal relations. Let us use $h(\cdot)$ to denote the corresponding predictor, and $\hat{\theta}$'s to denote the learned parameters without introducing any fairness constraint.

We summarize in Figure 3 the causal graph for the data generating process and the local causal modules that involve objectionable data generating components (denoted by red edges). Step 1 of Algorithm 2 sorts the sequence of nodes into $(A, C, X_1, X_2, X_3, X_4, Y)$. Then, we proceed by following this sequence and investigating each local causal module. When there is no additional knowledge or assumption on how one can directly modify the functional forms in Equation (5) to get a fair process, we illustrate step-by-step the application of the value instantiation rule (Algorithm 1) to decouple objectionable components by introducing reference points.

Consider an individual with original feature values $(a, x_1, x_2, x_3, x_4)$. In Figure 3(b), for the local causal module that has $X_2$ as the output, the direct parent nodes of $X_2$ include $(A, C, X_1)$, among which $A$ is the tail node of the red edge $A \rightarrow X_2$. We denote the reference point for $A$ with respect to the objectionable component $A \rightarrow X_2$ as $a|_{A \rightarrow X_2}^{\mathrm{ref}}$. Because of the existence of such objectionable

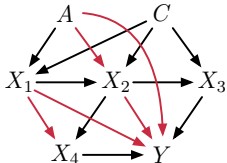 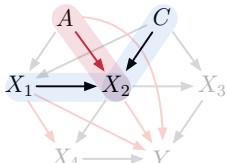 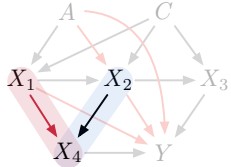 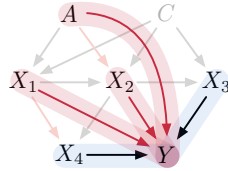

(a) Causal graph for data generating process

(b) Local causal module with $X_2$ as output

(c) Local causal module with $X_4$ as output

(d) Local causal module with $Y$ as output

Figure 3: Detailed illustrations of the application of value instantiation rule. Panel (a) presents the causal graph, with red edges denoting objectionable components. Panels (b) to (d) present local causal modules that involve objectionable component(s).

component, we use the reference point $a|_{A \to X_2}^{\text{ref}}$ as the input for $A$ since $A = \text{Tail}(A \to X_2)$, following Step 6 of Algorithm 1. We would like to note that $a|_{A \to X_2}^{\text{ref}}$ can be the same as or different from $a$, the original value of the variable $A$, as long as it is to the greatest benefit of the least benefit of the least advantaged individuals. For downstream causal modules that take $X_2$ as part of the inputs (Step 8 of Algorithm 1), the value to fill in the placeholder for $X_2$ should be:

$$\widehat{x}_2 = h_{X_2}(a|_{A \to X_2}^{\text{ref}}, c, x_1; \hat{\theta}_{X_2}). \tag{6}$$

For instance, although there is no objectionable component in the local causal module that has $X_3$ as the output, one should use $\widehat{x}_2$ as the input value of $X_2$ (since $X_2$ has ancestor node $A$ which was set to a reference point):

$$\widehat{x}_3 = h_{X_3}(c, \widehat{x}_2; \hat{\theta}_{X_3}). \tag{7}$$

Similarly in Figure 3(c), for the local causal module that has $X_4$ as the output, there is an objectionable component $X_1 \to X_4$. We can denote the reference point for $X_1$ with respect to the edge $X_1 \to X_4$ as $x_1|_{X_1 \to X_4}^{\text{ref}}$, and derive the predicted value for $X_4$ that decouples the objectionable component:

$$\widehat{x}_4 = h_{X_4}(x_1|_{X_1 \to X_4}^{\text{ref}}, \widehat{x}_2; \hat{\theta}_{X_4}), \tag{8}$$

where $x_1|_{X_1 \to X_4}^{\text{ref}}$ can equal to or differ from $x_1$, and $\widehat{x}_2$ is derived from Equation (6).

In Figure 3(d), for the local causal module that has $Y$ as the output, there are several objectionable components. We denote the reference point for $A$ with respect to the objectionable component $A \to Y$ as $a|_{A \to Y}^{\text{ref}}$, the reference point for $X_1$ with respect to $X_1 \to Y$ as $x_1|_{X_1 \to Y}^{\text{ref}}$, and the reference point for $X_1$ with respect to $X_2 \to Y$ as $x_2|_{X_2 \to Y}^{\text{ref}}$. One can derive the prediction for $Y$:

$$\widehat{y} = h_Y(a|_{A \to Y}^{\text{ref}}, x_1|_{X_1 \to Y}^{\text{ref}}, x_2|_{X_2 \to Y}^{\text{ref}}, \widehat{x}_3, \widehat{x}_4; \hat{\theta}_Y), \tag{9}$$

where $\widehat{x}_3$ and $\widehat{x}_4$ are derived from Equation (7) and Equation (8), respectively.

## C.2 REMARKS ON DERIVATION DETAILS

Firstly, if we compare the input values for $A$ in Equation (6) and Equation (9), we can see that the reference point values $a|_{A \to X_2}^{\text{ref}}$ and $a|_{A \to Y}^{\text{ref}}$ correspond to specific objectionable components instead of the variable $A$. We can also see that the input values for $X_1$ in Equation (8) and Equation (9) can be different, since $x_1|_{X_1 \to X_4}^{\text{ref}}$ and $x_1|_{X_1 \to Y}^{\text{ref}}$ correspond to objectionable components $X_1 \to X_4$ and $X_1 \to Y$, respectively. The potentially different reference point values taken by the same node, when it serves as the tail node for different objectionable components, quantitatively embody the discussion on the counter-factual analysis with respect to local causal mechanisms, instead of that with respect to variables only (Section B.1).

*Remark* C.1. The value of the reference point corresponds to the objectionable component, instead of the variable on which the reference point is assigned.

Secondly, the objectionable component, such as an edge in the causal graph, does not necessarily start from the protected feature or proxy variables, end at the final output, or represent a segment of a path starting from the protected feature and ending at the final output. In practical scenarios, the

discrimination can manifest itself at various locations in the data generating process. For example, the average height of a male is larger than that for a female. This is a neutral fact that does not involve discrimination based on the protected feature `sex`. However, if the data generating process mistreats individuals of a modest height and prefers taller individuals without any justifiable reason, the objectionable aspect in such process only starts from the variable `height` instead of the protected feature `sex`. In the worked-out example presented in Section C.1, we can see that the objectionable component $X_1 \rightarrow X_4$ starts from $X_1$ (not the protected feature $A$), ends at $X_4$ (not the final outcome $Y$), and is not a segment of any red path between $A$ and $Y$.

*Remark* C.2. The objectionable component is not limited to a segment of the causal path from the protected feature to the final outcome. Instead, it can represent any local causal mechanism responsible for problematic aspects in the data generation process.

Thirdly, in the data generating process, a single node can play different roles depending on the local causal module we focus on. For example, the protected feature $A$ receives reference point assignments in the local causal module that outputs $X_2$, as in Equation (6), or that outputs $Y$, as in Equation (9). However, in the local causal module that outputs $X_1$, since $A$ is not a tail node of any objectionable component, $A$ is not assigned a reference point. As another example, when $X_1$ serves as inputs for local causal modules, $X_1$ takes the original feature value $x_1$ when deriving $X_2$, as in Equation (6), but is assigned reference points $x_1|_{X_1 \rightarrow X_4}^{\text{ref}}$ and $x_1|_{X_1 \rightarrow Y}^{\text{ref}}$ when deriving $\widehat{x}_4$ and $\widehat{y}$, as in Equation (8) and Equation (9), respectively.

*Remark* C.3. The same node may have different roles in different local causal modules that take it as part of the inputs. The instantiated value should reflect the corresponding purpose of the node in the local causal module.

Lastly, if the objectionable component in the upstream causal modules were to behave in a neutral way, their outputs, which are inputs for the causal module of interest, would have been different from the observed feature values. For example, the local causal module that takes $X_3$ as the output involves $X_3$ and its direct parents $(C, X_2)$, i.e., $C \rightarrow X_3 \leftarrow X_2$. There is no objectionable component in this local causal module. However, in the upstream local causal module that takes $X_2$ as the output, as presented in Figure 3(b), $A \rightarrow X_2$ is an objectionable component and is counteracted by assigning the reference point $a|_{A \rightarrow X_2}^{\text{ref}}$ to $A$, as in Equation (6). Then, when deriving the value of $X_3$, had the objectionable component(s) in its own local causal module (in this example, none) as well as its upstream ones (in this example, it is $A \rightarrow X_2$) behaved in a neutral way, we obtain $\widehat{x}_3$ by utilizing $\widehat{x}_2$ instead of $x_2$ as in Equation (7). As another example, in Figure 3(d), for the local causal module that takes $Y$ as the output, $x_2|_{X_2 \rightarrow Y}^{\text{ref}}$ is utilized to fill in the placeholder for $X_2$ as an input in Equation (9), instead of $\widehat{x}_2$ or $x_2$. This is because the reference point $x_2|_{X_2 \rightarrow Y}^{\text{ref}}$ for $X_2$ is introduced to counteract the objectionable component $X_2 \rightarrow Y$, and it supersedes the derived value $\widehat{x}_2$ and the original feature value $x_2$.

*Remark* C.4. It is essential to keep track of the upstream of the inputs where reference points are assigned to their ancestor nodes, regardless of whether the local causal module of interest contains an objectionable component.

# D   EXPERIMENT DETAILS AND ADDITIONAL RESULTS

In this section, we provide our experimental details and present additional results. In Section D.1, we present implementation details of our framework and discuss the scalability of our approach. In Section D.2, we present the data generating process of the simulated data and summarize the derivation of prediction/decision of previous approaches that enforce causal fairness notions. In Section D.3, we provide experimental details and results on the real-world UCI Adult data set (Becker & Kohavi, 1996). In Section D.4, we present additional experimental results on the real-world Folktables data set (Ding et al., 2021).

## D.1   IMPLEMENTATION DETAILS AND SCALABILITY OF OUR APPROACH

We first present in Section D.1.1 implementation details of our framework that decouples objectionable data generating components to achieve procedural fairness. Then in Section D.1.2, we discuss the scalability of our approach.

Table 2: An illustrative comparison of computational costs to demonstrate the scalability of our approach. The causal model consists of 1,024 variables, and the average degree of the graph is 102, roughly $10\%$ of the number of variables.

| Computational costs | Our approach (forward pass) | Vanilla regressor (forward pass) |
|---|---|---|
| Number of parameters | $(7.56 \pm 0.04) \times 10^6$ | $7.66 \times 10^6$ |
| Multiplier-accumulator operations (MACs) | $(7.67 \pm 0.04) \times 10^6$ | $7.67 \times 10^6$ |

### D.1.1 THE IMPLEMENTAION DETAILS OF OUR FRAMEWORK

For the purpose of satisfying the requirements of procedural fairness (Section 2.2), our framework consists of two parts: the value instantiation rule in each local causal module (Section 4.2.1), and the configuration of reference point values (Section 4.2.2).

Given the causal model (in terms of a causal graph), we construct a neural network predictor for each local causal module between the node $V_i$ and its $d_{\text{in}}(V_i; \mathcal{G})$ direct parent nodes $\text{Parents}(V_i)$, such that: (i) the neural network predictor can be a classification or regression model, depending on the support of $V_i$; (ii) the neural network contains two hidden layers with a hidden dimension $\max\{5, d_{\text{in}}(V_i; \mathcal{G})\}$. We incorporate batch normalization (Ioffe & Szegedy, 2015) for each hidden layer and utilize the scaled exponential linear unit (SELU) activation function (Klambauer et al., 2017). The neural networks for local causal modules are optimized without any fairness constraint (Step 4 of Algorithm 2).

Within each local causal module, given the specification of objectionable data generating components, we decouple the objectionable components by keeping track of the appropriate value instantiation option for each input variable in $\text{Parents}(V_i)$ (Steps 5 – 10 of Algorithm 1). The exact value of reference points cannot be pre-determined before specifying the least advantaged individuals and actually carrying out the overall pipeline (Algorithm 2). Therefore, we use simulated annealing (Kirkpatrick et al., 1983) to derive the reference point configuration function $\text{ReferencePoint}(\cdot)$ when focusing on the least advantaged individuals, as summarized in Equation (4).

### D.1.2 THE SCALABILITY OF OUR APPROACH

Our framework can handle linear and nonlinear data generating processes, and scales well with the number of variables. We do not assume linearity of the data generating process when constructing predictive models for local causal modules, and the modeling of local causal modules is optimized without introducing any fairness constraint (Section D.1.1). To demonstrate the scalability of our approach, in addition to experimental details and results on simulated and real-world data sets (as we shall see in Sections D.2 – D.4), we provide a quantitative example of the computational cost of our approach for illustrative purposes.

Let us consider a scenario where there are 1,024 nodes in the causal graph DAG, with an average degree of 102 ($10\%$ of the number of nodes). In other words, the sum of in-degree (the number of direct parent nodes) and out-degree (the number of direct child nodes) of each node in the causal graph is on average 102. We randomly generate causal graphs that satisfy this configuration, and summarize in Table 2 the number of parameters and the number of multiplier-accumulator operations (MACs) during the inference phase, in comparison with a vanilla regression model. Our approach contains modeling of each local causal module in the causal graph (Section D.1.1), and the vanilla regressor is a neural network regression model that has two hidden layers with a hidden dimension of 2,300 (roughly twice the input dimension).

The decoupling of objectionable data generating components, which consists of propagating appropriate values according to the value instantiation rule and finding the reference point configuration $\text{ReferencePoint}(\cdot)$ that benefits the least advantaged individuals to the greatest extent possible, does not require retraining the classification or regression models for local causal modules. Therefore, the computational overhead in our approach only comes from the inference phase of the model during the simulated annealing process. No retraining is needed when the practitioner considers various reference point configurations, e.g., for different specifications of objectionable components and least

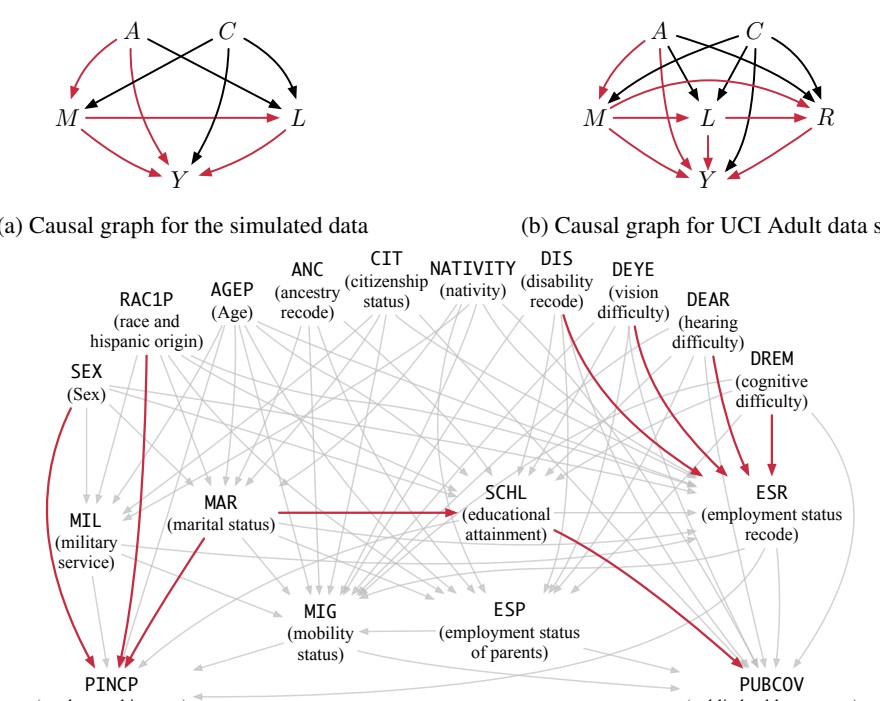

(a) Causal graph for the simulated data      (b) Causal graph for UCI Adult data set

(c) Causal graph for Folktables data set (`PUBCOV` prediction task)

Figure 4: Causal graph for experiments with simulated and real-world data sets.

advantaged individuals. As we can see from Table 2, the computational cost of the forward pass of our model is comparable to the vanilla regression model, indicating that our approach scales well with the number of variables in different practical scenarios.

### D.2 THE EXPERIMENT ON SIMULATED DATA

In Section 3, we present a linear model to illustrate *disguised procedural unfairness* resulting from the violation of Requirement I. In Section 5.1, we revisit the linear model to further illustrate the violation of Requirement II of procedural fairness. For the convenience of readers, we recap the linear data generating process and highlight the objectionable components in Figure 4(a).

$$
\begin{aligned}
A &\sim \text{Bernoulli}(p_A), \\
C &= \epsilon_C, \\
M &= \theta_A^M A + \theta_C^M C + \theta^M + \epsilon_M, \\
L &= \theta_A^L A + \theta_C^L C + \theta_M^L M + \theta^L + \epsilon_L, \\
Y &= \theta_A^Y A + \theta_C^Y C + \theta_M^Y M + \theta_L^Y L + \theta^Y + \epsilon_Y.
\end{aligned}
\tag{10}
$$

Equation (10) describes the linear data generating process, with the objectionable components highlighted by parameters in red. We would like to note that the correspondence between the objectionable components, i.e., red edges in Figure 4(a), and the red parameters in Equation (10) results from the linear model, and that such component-parameter correspondence is not available in general scenarios, as we discussed in Section 4.1.

Various causal fairness notions have been proposed in the previous literature (Section A.1). In our experiment on the simulated data, we consider different constraints on parameter optimization, including *No Unresolved Discrimination* proposed by Kilbertus et al. (2017), and *Fair Inference on Outcome* considered in Nabi & Shpitser (2018) and follow-up works (Nabi et al., 2019; 2022). We

utilize the linear regression model without fairness constraints as the baseline, and evaluate linear models when different causal fairness constraints are enforced. Kilbertus et al. (2017) require $\hat{\theta}_A^Y = 0$ and $\hat{\theta}_M^Y + \hat{\theta}_L^Y \cdot \hat{\theta}_M^L = 0$, which do not yield a unique set of fitted parameters. Therefore, we consider two different sets of fitted parameters in Figure 1(b), both of which satisfy the fairness constraints. Nabi & Shpitser (2018) require $\hat{\theta}_A^Y = 0$ and $\hat{\theta}_A^M = 0$ as the sufficient condition to satisfy the causal fairness requirement, and we present the fitted parameters in Figure 1(c).

In Figure 2(a), we present the summary of decision-making results. We consider the baseline linear regression model, two linear models that satisfy *No Unresolved Discrimination* (Kilbertus et al., 2017), and an additional linear model that follows *Fair Inference on Outcome* (Nabi & Shpitser, 2018; Nabi et al., 2019; 2022). Specifically, to enforce *No Unresolved Discrimination* (Kilbertus et al., 2017), we utilize the fitted parameters with fairness constraints in the linear regression model to derive the continuous prediction $\widehat{Y}$. To deploy *Fair Inference on Outcome* (Nabi & Shpitser, 2018; Nabi et al., 2019; 2022), we follow the inference approach via box constraints (Nabi & Shpitser, 2018), draw Monte Carlo samples for intermediate variables $(M, L)$ and integrate over intermediate linear regression outputs to derive the prediction $\widehat{Y}$.

To output a binary decision from the continuous predicted value $\widehat{Y}$ of linear models, we apply certain thresholds. Any value exceeding the threshold is classified as the favorable decision, such as loan approval or program acceptance. The threshold reflects the resource abundance: a higher threshold indicates scarcer resources, resulting in fewer favorable decisions distributed. We are interested in the potential situation improvement for the least advantaged individuals, and compare the decisions made by different models, including the baseline model and the ones that enforce causal fairness notions.

In this illustrative example, among those that are rejected by all decision-making policies, the disadvantaged individuals proportionally suffer more; among those that are accepted by all decision-making policies, the disadvantaged individuals proportionally prosper less. For instance in Figure 2(a), let the threshold be $0.6$, i.e., the favorable decision is assigned when the predicted value satisfies $\widehat{Y} > 0.6$. Among those that are rejected by all models, the disadvantaged group consists of $\frac{29.3\%}{29.3\%+37.5\%} = 43.9\%$, which is larger than the marginal group representation $40.0\%$. However, if we set the threshold to $0.3$ and assign the favorable decision as long as the predicted value satisfies $\widehat{Y} > 0.3$, the disadvantaged group only consists of $\frac{27.2\%}{27.2\%+54.5\%} = 33.3\%$ ($< 40.0\%$) among those that are accepted by all models. This indicates that even if causal fairness notions are imposed, disadvantaged individuals disproportionately face adverse outcomes, demonstrating the violation of requirements for procedural fairness.

### D.3 THE EXPERIMENT ON UCI ADULT DATA SET

In this section, we provide experimental details and additional results on the real-world UCI Adult data set (Becker & Kohavi, 1996). The data set contains 14 feature variables and 1 target variable. The features include individual's personal record attributes, e.g, `age`, `race`, `sex`, `marital status`, `native country`, individual's social and educational record attributes, e.g., `education level`, `relationship to household`, and individual's occupation related attributes, e.g., `class of work`, `occupation`, `work hours per week`. The task is to predict whether an individual has an annual `income` that exceeds USD 50,000.

Following Nabi & Shpitser (2018) and Chiappa (2019), we model the data generating process in Figure 4(b), where following variables are included: $A$ for `sex`, $C$ for the tuple (`age`, `native country`), $M$ for `marital status`, $L$ for `education level`, $R$ for the tuple (`class of work`, `occupation`, `work hours per week`), and $Y$ for `income`. The variables `race`, `capital gain/loss` are omitted. We utilize the default train-test data split, and use 32,561 and 16,281 records for training and testing purposes, respectively.

In Figure 4(b), we highlight in red the problematic aspects defined in the previous literature (Chiappa, 2019). Specifically, Chiappa (2019) would like to remove the direct effect $A \rightarrow Y$, as well as the effect of $A$ on $Y$ through $M$, namely, along paths $A \rightarrow M \rightarrow \cdots \rightarrow Y$. Different from previous approaches that enforce fairness constraints on model parameters (Kilbertus et al., 2017; Nabi & Shpitser, 2018; Nabi et al., 2019; 2022), Chiappa (2019) proposes to reconstruct the variables that are descendants of the protected feature $A$ along problematic pathways. Chiappa (2019) introduces

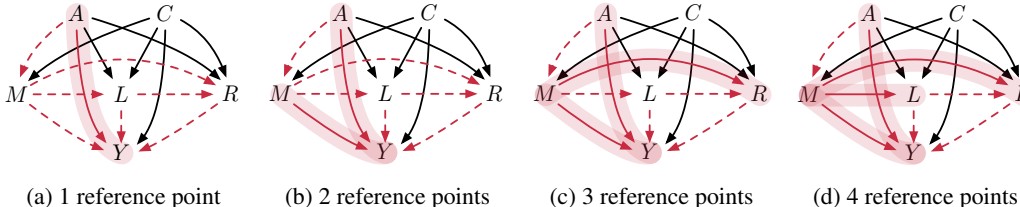

|          |          |          |          |
| :------: | :------: | :------: | :------: |
| (a) 1 reference point | (b) 2 reference points | (c) 3 reference points | (d) 4 reference points |

Figure 5: Different objectionable component configurations in experiments on the UCI Adult data set. Red edges denote potential locations of objectionable components, among which solid-line edges (highlighted with contours) represent objectionable components that we decouple through reference points, and dashed-line edges represent potentially neutral (not objectionable) components.

additional latent variables $(H_M, H_L, H_R)$ for intermediate variables $(M, L, R)$, and utilizes the latent inference-projection approach to "correct" descendants of $A$. The predicted outcome $\widehat{Y}$ is derived by sampling from the fitted model likelihood, which encapsules Monte Carlo samples from Gaussian approximations of latents' posterior distributions given observed features (Chiappa, 2019). The comparison between baseline prediction and fair (in terms of *Path-Specific Counterfactual Fairness*, Chiappa 2019) prediction is summarized in the first column of each subplot in Figure 2(c).

Our framework utilizes reference points to decouple objectionable components in the data generating process of the predicted outcome. We view red edges specified by previous literature in Figure 4(b) as potential locations to assign reference points, and consider different configurations of reference points. The number of all possible combinations of objectionable components increases exponentially with their potential locations. For example, for $k_0$ potentially objectionable edges, there are $\sum_{k_1=0}^{k_0} \binom{k_0}{k_1} = 2^{k_0}$ different configurations of actually objectionable components. Therefore, for the purpose of presenting a meaningful comparison with the previous approach (Chiappa, 2019), we did not present the exhaustive list of all reference point configurations, and specifically pay attention to objectionable components that involve variables $(A, M, Y)$.

Our framework differs from previous causal fairness notions (including *Path-Specific Counterfactual Fairness* proposed by Chiappa 2019) and conducts counter-factual analysis with respect to local causal mechanisms (Section B.1). The value of reference point is not a pre-specified quantity readily available in an *ex ante* way. By "*ex ante*", we are referring to the common starting point for counterfactual causal inference, for example, reasoning about the situation were a female individual male, i.e., flipping sex from female to male, while keeping other observed features unchanged (Chiappa, 2019). In contrast, the reference points are derived in an *ex post* way by solving an optimization problem characterized in Equation (4).

For Figure 2(c) presented in Section 5.2, we utilize simulated annealing to determine reference point values that maximize benefit for the least advantaged individuals, who, in this context, are the female group. Specifically, we present causal graphs when the set of objectionable components contains 1 reference point $\{A \rightarrow Y\}$ as in Figure 5(a), 2 reference points $\{A \rightarrow Y, M \rightarrow Y\}$ as in Figure 5(b), 3 reference points $\{A \rightarrow Y, M \rightarrow Y, M \rightarrow R\}$ as in Figure 5(c), and 4 reference points $\{A \rightarrow Y, M \rightarrow Y, M \rightarrow R, M \rightarrow L\}$ as in Figure 5(d). We provide in Table 3 detailed reference point values as well as improvements in approval rates (i.e., favorable decision for predicted income higher than USD 50,000) for different groups, compared to the unconstrained baseline, where Cases (i) – (iv) correspond to Figures 5(a) – 5(d).

If we compare Case (i) with Case (ii) and Case (iii) in Table 3, we can see that the reference point configuration that maximizes the benefit of the least advantaged individuals does not necessarily involve treating disadvantaged individuals (female) as if they were advantaged (male). For example, in Case (ii), when both $A \rightarrow Y$ (from sex directly to income) and $M \rightarrow Y$ (from marital status directly to income) are objectionable components, as presented in Figure 5(b), we should set "female" and "married" as reference points for inputs to corresponding objectionable components, and carry out the same prediction derivation procedure (Algorithm 2) for everyone. In other words, procedural fairness requires us to decouple objectionable components by treating all individuals as

Table 3: Compare the effectiveness of different reference point configurations for the purpose of decoupling objectionable components presented in Figure 5.

| Case | Objectionable component(s) | Reference point configuration(s) | Ground-truth `income` | Approval rate for `female` | Approval rate for `male` |
|------|------|------|------|------|------|
| (i) | $A \to Y$ | $a\|_{A \to Y}^{\mathrm{ref}} = $ `male` | `low` | $+0.0059$ | $+0.0008$ |
| | | | `high` | $+0.0286$ | no change |
| (ii) | $A \to Y$ | $a\|_{A \to Y}^{\mathrm{ref}} = $ `female` | `low` | $+0.3870$ | $+0.3233$ |
| | $M \to Y$ | $m\|_{M \to Y}^{\mathrm{ref}} = $ `married` | `high` | $+0.4494$ | $+0.2693$ |
| (iii) | $A \to Y$ | $a\|_{A \to Y}^{\mathrm{ref}} = $ `female` | `low` | $+0.3911$ | $+0.3277$ |
| | $M \to Y$ | $m\|_{M \to Y}^{\mathrm{ref}} = $ `married` | `high` | $+0.4416$ | $+0.2714$ |
| | $M \to R$ | $m\|_{M \to R}^{\mathrm{ref}} = $ `married` | | | |
| (iv) | $A \to Y$ | $a\|_{A \to Y}^{\mathrm{ref}} = $ `male` | `low` | $+0.4828$ | $+0.4627$ |
| | $M \to Y$ | $m\|_{M \to Y}^{\mathrm{ref}} = $ `married` | `high` | $+0.4260$ | $+0.2978$ |
| | $M \to R$ | $m\|_{M \to R}^{\mathrm{ref}} = $ `married` | | | |
| | $M \to L$ | $m\|_{M \to L}^{\mathrm{ref}} = $ `single` | | | |

if they were females along edges $A \to Y$, and married along $M \to Y$. This is different from the common starting point in previous causal fairness notions, where the focus is on the certain causal effect on outcome $Y$ if `sex` variable $A$ were to be flipped from female to male (Kilbertus et al., 2017; Nabi & Shpitser, 2018; Chiappa, 2019; Wu et al., 2019; Nabi et al., 2019; 2022).

### D.4 THE EXPERIMENT ON FOLKTABLES DATA SET (PUBCOV PREDICTION TASK)

In this section, we present experimental results on the real-world Folktables data set (Ding et al., 2021). In particular, we consider the prediction task for public health coverage, where there are 17 features and the target variable is PUBCOV. We retrieve ACS PUMS data for the year 2021, and consider the following states: CA, FL, and NY.

We assume that the causal model for the data generating process among variables of interest can be presented as in Figure 4(c). The detailed information about the possible values of variables can be found in the document on data dictionary from the US Census Bureau (Bureau, 2021). Following Ding et al. (2021), we preprocess the data and consider individuals with the age less than 65, and the annual income no more than USD 30,000. The focus of this prediction task is on the public health coverage for low-income individuals. After preprocessing, we perform data splits and get training and testing sets for CA (99,840 for training, 33,281 for testing), FL (50,134 for training, 16,712 for testing), and NY (48,212 for training, 16,071 for testing), respectively.

In Figure 4(c), we use red edges to denote objectionable data generating components and light-gray edges to denote neutral components. We consider the group of the least advantaged individuals among low-income individuals, and focus specifically on those having a disability and/or experiencing vision, hearing, or cognitive difficulties.

We summarize in Table 4 the derived reference point configurations for each state among CA, FL, and NY. With the same causal graph and specifications of objectionable data generating components, and the shared criterion for the least advantaged group characteristics, we can observe that different states may require different sets of reference point configurations to satisfy requirements of procedural fairness. For instance, for the objectionable component SEX $\to$ PINCP (the edge from `sex` to `total annual income`), the reference point value yields `male` in CA and NY, but `female` in state FL. We can also observe shared patterns of reference point configurations among different

Table 4: The summary of reference point configurations for the public health coverage prediction task in Folktables data set (Ding et al., 2021), among states `CA`, `FL`, and `NY`. The reference point values of variables are recorded according to the data dictionary document from the US Census Bureau (Bureau, 2021).

| Objectionable component | CA | FL | NY |
|---|---|---|---|
| ReferencePoint(SEX → PINCP) 
 Tail node: SEX (sex) | male | female | male |
| ReferencePoint(RAC1P → PINCP) 
 Tail node: RAC1P (race & hispanic origin) | some other race alone | white | white |
| ReferencePoint(MAR → PINCP) 
 Tail node: MAR (marital status) | married | separated | divorced |
| ReferencePoint(MAR → SCHL) 
 Tail node: MAR (marital status) | never married or under 15 | separated | married |
| ReferencePoint(SCHL → PUBCOV) 
 Tail node: SCHL (educational attainment) | Grade 5 | Grade 7 | Grade 5 |
| ReferencePoint(DIS → ESR) 
 Tail node: DIS (disability status) | with | with | w/out |
| ReferencePoint(DEYE → ESR) 
 Tail node: DEYE (vision difficulty) | w/out | with | with |
| ReferencePoint(DEAR → ESR) 
 Tail node: DEAR (hearing difficulty) | w/out | w/out | w/out |
| ReferencePoint(DREM → ESR) 
 Tail node: DREM (cognitive difficulty) | with | with | with |

states. For instance, let us compare the objectionable components DREM → ESR (the edge from `cognitive difficulty` to `employment status`) and DEAR → ESR (the edge from `hearing difficulty` to `employment status`). The reference point configuration specifies that in order to receive a favorable prediction/decision, the individual characteristics need to be `with` cognitive difficulty but `without` hearing difficulty. This indicates that in the data generating process for determining public health coverage across considered states, the objectionable aspects are more lenient (in terms of the level of discrimination) towards cognitive difficulties (DREM), yet demonstrate more severity in the discrimination against individuals with hearing difficulty (DEAR).

## E    FURTHER DISCUSSIONS

In this section, we present further discussions on the comparison between outcome and procedural emphases of algorithmic fairness (Section E.1), implications of our results on procedural fairness (Section E.2), and potential limitations and future works (Section E.3).

### E.1    ALGORITHMIC FAIRNESS: OUTCOME EMPHASIS VS. PROCEDURAL EMPHASIS

In this work, motivated by the procedural conceptions of justice (Rawls, 1971; 2001), we focus on procedural fairness of the data generating process itself for prediction or decision-making. Rawls's full statement involves lexically ordered principles of justice, as well as accompanying priority rules (Rawls, 1971; 2001). The primary object of interest in Rawls's theory of justice is the construction of the structure of social institutions, and the scope of consideration goes beyond algorithmic fairness with respect to a data generating process. That being said, we view the automated decision-making as a microcosm of social institutions, and believe it is valuable to borrow the wisdom from the

rich literature of political philosophy and moral theories, in particular, Rawls's advocacy for *pure procedural justice*, to carefully consider the requirements and implications of procedural fairness.

The difficulty of specifying a standalone criterion for the just or fair outcome in general scenarios is not surprising. Previous literature has proposed various fairness notions defined on predicted outcome or decision, but not all of them are compatible with each other (Chouldechova, 2017; Kleinberg et al., 2017). One can observe different public opinions regarding which notion we should use to define the "fair" predicted outcome or decision (Saxena et al., 2019). There are also debates in moral and political philosophy literature over how one should define the proper space in which equality is desirable (Cohen, 1989; Anderson, 1999). Therefore, motivated by *pure procedural justice*, by characterizing the property of data generating process and further making sure that such procedure is actually carried out in an impartial way, we propose the framework to decouple the objectionable data generating components, in order to achieve procedural fairness.

## E.2 IMPLICATION OF OUR RESULTS ON PROCEDURAL FAIRNESS

In Section B, we discussed the differences and connections of our framework compared to previous works. In Section C, we illustrated in detail the overall pipeline as well as the derivation detail of our framework. In Section D, we presented experimental details and additional results to demonstrate the effectiveness of our approach. To provide a more complete picture of the implication of our results on procedural fairness, we extend our discussion in Section B.1 on the distinction between the counter-factual analysis on variables and that on local causal mechanisms, and further reflect on the goal of procedural fairness.

### E.2.1 THE VERSATILE NATURE OF OUR FRAMEWORK

Our framework of procedural fairness through decoupling objectionable data generating components is versatile, capable of accommodating various configurations while still remaining principled.

To begin with, our framework is versatile with respect to the configuration of objectionable components. If additional knowledge or assumption is available in terms of how one should modify or correct the behavior of local causal modules, our framework can immediately make use of them on corresponding objectionable data generating components (Step 2 of Algorithm 1). When there are multiple configurations of objectionable components over potential locations, our framework can readily accommodate different options and provide reference points that satisfy requirements of procedural fairness, as we have seen in our implementation details (Sections D.1.1 – D.1.2) and experimental results (Section 5.2 and Sections D.2 – D.4).

In addition, our framework is also versatile with respect to the specification of least advantaged individuals. In our framework, we do not make specific distinctions between "protected features" and "regular features" when decoupling objectionable components for procedural fairness. In practical scenarios, objectionable components can manifest itself in various forms and locations in the data generating process. This necessitates counter-factual analyses with respect to local causal mechanisms instead of variables (Section B.1). Our framework also adapts to more precise definitions of the "least advantaged individuals", which are more fine-grained than group categorizations only by reference to values of the protected feature. Previous literature has recognized the shortcomings of defining disadvantaged individuals only in terms of certain protected features, considering one protected feature at a time (Crenshaw, 1990; Kearns et al., 2018; Foulds et al., 2020; Kong, 2022). Our framework can readily adapt to more precise definitions of the "least advantaged individuals" or a specific individual, and find reference points that satisfy the requirements of procedural fairness (Sections D.3 – D.4).

Furthermore, our framework is extendable and readily adapts to scenarios where new variables and causal mechanisms are incorporated into the system. We consider objectionable components in each local causal module (Algorithm 1), and then incorporate reference points and aggregate local causal modules to derive the final prediction (Algorithm 2). When additional variables and/or causal mechanisms are introduced, our framework can be naturally extended to address procedural fairness in the new system.

### E.2.2   TOWARDS A TRANSPARENT FRAMEWORK FOR PROCEDURAL FAIRNESS

In our framework, the decoupling of objectionable components is operated in a localized manner. In particular, we address procedural fairness at the level of the causal influence between a variable and its direct causes. This is more straightforward than attempting to decipher causal effects from the protected feature $A$ to the outcome $Y$, especially when there are multiple paths between them and other variables are encompassed on the paths. The utilization of reference points in our framework provides a transparent view of the bias mitigation strategy, making it easier to understand how procedural fairness is enforced.

It has been recognized in algorithmic fairness literature that the fairness pursuit is not a purely technical problem (O'neil, 2017; Kearns & Roth, 2019; Barocas et al., 2019). The collaboration between algorithm designers and domain experts, such as ethical committees and social scientists, is of vital importance. We believe a transparent framework for procedural fairness is crucial for incorporating feedback and continuously improving the prediction or decision-making system.

### E.2.3   NOT IN CONFLICT WITH CAUSAL FAIRNESS NOTIONS

Causal fairness notions provide quantification tools to evaluate the influence from the protected feature $A$ (or related proxy variables) to the final output $Y$ or its prediction $\widehat{Y}$ along certain paths. Constraining causal effects along objectionable paths alone does not give us a clear guidance on what to expect for causal effects along other neutral paths.

Although enforcing causal fairness notions with existing proposals can result in *disguised procedural unfairness*, our framework does not necessarily conflict with previous causal fairness notions. We share the intuition with previous literature that causal reasoning is essential when addressing algorithmic fairness with procedural emphasis (Kilbertus et al., 2017; Kusner et al., 2017; Nabi & Shpitser, 2018; Chiappa, 2019; Wu et al., 2019; Nabi et al., 2019; 2022).

Incorporating the focus on procedural fairness requirements in our framework, a potential option to address *disguised procedural unfairness* with previous causal fairness notions could be: in addition to imposing causal fairness constraints exclusively on objectionable paths, consider also causal effects along all other neutral paths, and ensure their proximity to causal effects in the scenario without the aforementioned causal fairness constraints enforced in the first place. Additional technical challenges may arise for previous causal fairness approaches, such as exhaustively listing all paths for causal effect estimation or bounding, evaluating causal effect identification conditions, and determining the proper causal estimands.

### E.3   POTENTIAL LIMITATIONS AND FUTURE WORKS

In this work, we explore how procedural guarantees can be implemented on data generating process itself to achieve procedural fairness. Our framework rests upon causal modularity (Spirtes et al., 1993; Pearl, 2009), also known as exogeneity (Engle et al., 1983) and independence of causal mechanism (Peters et al., 2017), utilizing reference points together with appropriate value instantiation rule to decouple objectionable data generating components from neutral ones. In certain systems, when there is a limited number of observed variables, or there is a lack of guidance on what constitutes objectionable components, our framework may not be effective. However, the versatile and transparent nature of our approach (Section E.2.1, Section E.2.2) enables the potential for adaptive and extendable improvements, when further information about the system is available.

Going beyond static settings, the Markov condition and its variants (Lauritzen, 1996; Spirtes et al., 1993; Richardson, 2003) may not hold true in dynamic settings or in causal systems that involve feedback loops represented with a directed cyclic graph (DCG) (Spirtes, 1995). As a result, causal modularity may not hold true, and our framework cannot be directly applied to decouple objectionable data generating components. Incorporating the interplay among various data generating processes and ensuring procedural fairness in long-term and dynamic settings remains an important question, and we leave them for future research.

