# OpenReview forum: "Procedural Fairness Through Decoupling Objectionable Data Generating Components"
_ICLR.cc/2024/Conference — ICLR 2024 spotlight_

### Official Review · Reviewer_R8GC · 2023-10-25

**Soundness:** 3 good
**Presentation:** 3 good
**Contribution:** 2 fair
**Rating:** 6
**Confidence:** 2

**Summary:**

The authors look at the issue of disguised procedural unfairness. They focus on the data generation process and try and decouple parts of the process which could be objectionable. Through their findings, they advocate for procedural fairness in the data generation process and argue that just trying to fit parameters in a fair manner may not be viable.

**Strengths:**

The authors do a good job of situating their work in terms of other efforts in the fairness literature. It is an important issue and preventing disguised procedural unfairness is an area where we need to better understand the practical harms as a community.

**Weaknesses:**

The paper brings in a number of areas (e.g. causal modeling, procedural fairness, procedural justice, hypothesis classes, graphical models, etc.) making it challenging to parse. There are only two experiments, one of which is a synthetic dataset. The algorithms are difficult to parse: examples, digressions, and new notation and comments are mixed in with math. The heatmap experiments embed several concepts, references and notation, making it difficult to parse their claims.

**Questions:**

- Does your framework require a causal model between attributes?
- Are their simpler toy examples which you can illustrate as a warm-up?
- In practice, what does it mean to constrain the data generating process (DGP)? Are we essentially filtering out data based on constraints before learning a model?
- Are there other examples of work which look at the DGP and constraining it? Why haven't they?

---

> ### Author Response · Authors · 2023-11-20
> **Response to Comments by Reviewer R8GC**
>
> Thank you for the thoughtful questions and comments, and for the time devoted! Below please see our responses to specific points in the review comments:
>
> ---
>
> ### **C1:** "There are only two experiments, one of which is a synthetic dataset."
>
> **R1:** Thanks for the comment. In addition to the two sets of experiments on simulated data and the UCI Adult data set, we have included additional experimental results on the real-world Folktables data set (Ding et al., 2021). In particular, we consider the public health coverage prediction task where there are 17 features and 1 target variable. In light of the comment, we have included the additional experimental results in Appendix D.4, along with the side note `Re: C1 by Reviewer R8GC` on page 23.
>
> ---
>
> ### **Q2:** "Does your framework require a causal model between attributes?"
>
> **R2:** Thanks for the question on the causal model. Yes, our framework utilizes the causal model between attributes. Our assumption on the availability of the causal model (e.g., in terms of a causal graph) is consistent with previous literature on causal fairness. For the convenience of locating the related material, we provide the side note `Re: Q2 by Reviewer R8GC` on page 5.
>
> ---
>
> ### **Q3:** "Are their simpler toy examples which you can illustrate as a warm-up?"
>
> **R3:** If we understood it correctly, the question is about the illustrative example of our framework. Yes, we presented a worked-out example in Appendix C to illustrate in detail Algorithm 1 and Algorithm 2 of our framework. For the convenience of locating the related material, we provide the side note `Re: Q3 by Reviewer R8GC` on page 21. Please kindly let us know if we misunderstood the question.
>
> ---
>
> ### **Q4:** "In practice, what does it mean to constrain the data generating process (DGP)? Are we essentially filtering out data based on constraints before learning a model?"
>
> **R4:** Thanks for the question. By "constraint" on DGP we are referring to the fairness constraints on certain causal quantities proposed in previous causal fairness literature. We are not filtering out data based on such constraints before learning a model. Instead, previous causal fairness notions propose that the causal effect of interest should be bounded. For completeness, we presented the literature review on causal fairness notions in Appendix A.1. We provide the side note `Re: Q4 by Reviewer R8GC` on page 17.
>
> ---
>
> ### **Q5:** "Are there other examples of work which look at the DGP and constraining it? Why haven't they?"
>
> **R5:** Thanks for the question about previous causal fairness notions. Yes, there are previous works that look at DGP and constrain it. For instance, Counterfactual Fairness (Kusner et al., 2017) requires that the counterfactual causal effect between the protected feature and the prediction should be bounded by a small positive constant. We presented the literature review on causal fairness notions in Appendix A.1, and provided discussions on differences and connections of our approach compared to previous causal fairness works in Appendix B.1. We provide the side note `Re: Q5 by Reviewer R8GC` on page 19.
>
> ---
>
> ### **Reference**
>
> Frances Ding, Moritz Hardt, John Miller, and Ludwig Schmidt. Retiring adult: New datasets for fair machine learning. In _Advances in Neural Information Processing_ Systems, volume 34, pp. 6478–6490, 2021.
>
> Matt Kusner, Joshua Loftus, Chris Russell, and Ricardo Silva. Counterfactual fairness. In _Advances in Neural Information Processing Systems_, pp. 4066–4076, 2017.

---

> > ### Comment · Reviewer_R8GC · 2023-11-21
> >
> > We appreciate the clarifications offered and have increased our score.

---

> > > ### Author Response · Authors · 2023-11-21
> > > **Thank Reviewer for the Acknowledgment**
> > >
> > > Thank you for letting us know that our clarifications are helpful! We are extremely grateful for the thoughtful questions and comments, and for the encouraging acknowledgment. We strive to make our work clear, transparent, and informative. Please let us know if you would like to suggest any further changes.

---

### Official Review · Reviewer_GSnz · 2023-10-29

**Soundness:** 3 good
**Presentation:** 2 fair
**Contribution:** 2 fair
**Rating:** 6
**Confidence:** 4

**Summary:**

The paper addresses the issue of objectionable components in a data-generating process in predictive modeling that might lead to unfair predictions. The focus is on scenarios where model parameters exhibit objectionable aspects, and seeks to isolate and correct these aspects to achieve procedural fairness. The proposed approach integrates concepts from causal inference, fairness constraints, and optimization to propose algorithms aimed at achieving fairer outcomes. It is based on two main requirements: (1) Fair Equality of Opportunity: The opportunity should be open and attainable, with the same prospects of success, for those who are at the same level of talent and ability, and have the same willingness to use them. (2) The Difference Principle: The (social and economic) inequalities are to be arranged so that they are to the greatest benefit to the least advantaged members of the society.

**Strengths:**

The approach is well-motivated and well-presented.
The addressed problem is interesting.
The authors provide a systematic examination and manipulation of objectionable components, and the use of reference points and value instantiation rules, with the goal to mitigate unfairness in predictive modeling stemming from objectionable data-generating processes.

**Weaknesses:**

The approach is motivated by very simple examples/models involving causal dependencies.
For more complex models, it requires a deep understanding of causal relationships within the data, which may require expert knowledge and extensive data analysis.
The approach builds on the identification of objectionable components, the availability of causal relations, and the correct specification of local causal modules (which is extremely hard for real-world datasets and scenarios). Moreover, in real-world applications, the distinction between objectionable and neutral components might not be possible. Is there a practical way to identify objectionable and neutral components in high-dimensional settings with intricate and unknown dependencies between variables?
While the optimization problem of configuring reference point values to maximize the benefits for the least advantaged individuals makes sense, it is unclear how complex it is.
The evaluation is based on the UCI Adult dataset only, and the results might not be generalizable across different domains or datasets. Also, a comparison to other fairness strategies might be insightful.

**Questions:**

Please see comments above.

**Details Of Ethics Concerns:**

None.

---

> ### Author Response · Authors · 2023-11-20
> **Response to Comments by Reviewer GSnz**
>
> Thanks for the thoughtful and detailed comments, as well as the time and effort devoted! Below please see our responses to specific comments and questions:
>
> ---
>
> ### **Q1:** "Is there a practical way to identify objectionable and neutral components in high-dimensional settings with intricate and unknown dependencies between variables?"
>
> **R1:** Thanks for the thoughtful question about the availability of the causal model and the specification of objectionable and neutral components! With the recent developments in the causal discovery literature, under certain mild assumptions, the causal graph can be recovered from high-dimensional settings. Since causal discovery (e.g., to find the causal graph) and algorithmic fairness (to define/quantify discrimination given a causal model) are different tasks, we follow previous causal fairness literature and assume the availability of the causal graph. It has been recognized in the literature that the fairness pursuit is not a pure technical question, and that it requires expert knowledge from different domains. Therefore, our framework takes the specified objectionable and neutral components, together with the causal graph, as part of the input. We reveal and address the often-overlooked _disguised procedural unfairness_, and show that even if the causal graph is available, previous causal fairness notions can still violate the requirements of procedural fairness.
>
> ---
>
> ### **C2:** "While the optimization problem of configuring reference point values to maximize the benefits for the least advantaged individuals makes sense, it is unclear how complex it is."
>
> **R2:** Thanks for carefully considering the detail of our approach! In addition to our code implementation in the supplementary material (we provided [the anonymous link](https://anonymous.4open.science/r/DecouplingObjectionable) in the paper), we have also included descriptions of implementation details in Appendix D.1.1, and the illustration (with more than 1000 variables) and discussion on scalability of our approach in Appendix D.1.2. For the convenience of locating the related material, we provide the side note `Re: C2 by Reviewer GSnz` on page 24.
>
> ---
>
> ### **C3:** "The evaluation is based on the UCI Adult dataset only, and the results might not be generalizable across different domains or datasets."
>
> **R3:** Thanks for the question about experimental results on other data sets. In light of your comment, we have included additional experimental results on the real-world Folktables data set (Ding et al., 2021). We consider the public health coverage prediction task, and present the additional experimental results in Appendix D.4, along with the side note `Re: C3 by Reviewer GSnz` on page 28.
>
> ---
>
> ### **C4:** "Also, a comparison to other fairness strategies might be insightful."
>
> **R4:** Thanks for sharing the suggestion. We completely agree, and this is exactly why we presented literature review over different fairness strategies, including causal fairness notions (Appendix A.1), scenarios with responsive agents (Appendix A.2), and fair representation learning (Appendix A.3). We also discussed in detail differences and connections of our approach compared to previous fairness strategies (Appendix B) and summarized the comparison in Table 1. For the convenience of locating the related material, we provide the side note `Re: C4 by Reviewer GSnz` on page 17.
>
> ---
>
> ### **Reference**
>
> Frances Ding, Moritz Hardt, John Miller, and Ludwig Schmidt. Retiring adult: New datasets for fair machine learning. In _Advances in Neural Information Processing_ Systems, volume 34, pp. 6478–6490, 2021.

---

> ### Author Response · Authors · 2023-11-21
> **Eagerly Looking Forward to Feedback on Our Response and Revised Manuscript**
>
> We are very grateful for your thoughtful and detailed comments, as well as the time and effort devoted!
>
> Following your constructive suggestions, we have incorporated following changes in our revision:
>
> 1. We include **descriptions of our implementing details**, which serve as complements to our code implementation provided in the supplementary material.
>
> 1. We provide a quantitative demonstration of the **scalability of our framework**. We consider a scenario involving more than 1,000 variables to illustrate the computational costs.
>
> 1. We include **additional experimental results** on the real-world data on the public healthcare prediction task.
>
> 1. We provide the pointer to the **comparison to other fairness strategies**, including the literature review and the detailed discussions on differences and connections.
>
> We have also provided color-coded side notes in the revised manuscript to help locate the related material and discussions.
>
> We are very happy to get to know that you found our approach "well-motivated and well-presented", and that the problem we address "is interesting". We sincerely hope our further clarifications can fully address the questions on our framework implementation, and can be helpful in the evaluation of our results. We are eagerly looking forward to your kind feedback.

---

> > ### Comment · Reviewer_GSnz · 2023-11-22
> > **Thank you for the revision.**
> >
> > I thank the authors for their response and the consideration of my questions in the revision of the paper. Although I believe there is still room for improvement in the experimental analysis, the authors have satisfactorily addressed several of my primary concerns. Therefore, I will raise the score.

---

> > > ### Author Response · Authors · 2023-11-22
> > > **Thank Reviewer for the Acknowledgment**
> > >
> > > Thanks again for the constructive, thoughtful, and detailed comments and suggestions, which helped us improve our work! We are very glad that our revision and response addressed your primary concerns. We are extremely grateful for the time devoted, and for the encouraging acknowledgement. We try our best to make our work clear, precise, and informative. Please feel free to let us know if you would like to suggest any changes to further enhance our work.

---

### Official Review · Reviewer_2K3E · 2023-11-06

**Soundness:** 4 excellent
**Presentation:** 3 good
**Contribution:** 3 good
**Rating:** 8
**Confidence:** 3

**Summary:**

The paper aim to address inadvertent biases in the data generation process that can affect even neutral aspects of this process, potentially compromising fairness.
This is referred to as procedural unfairness. The authors propose a framework to decouple objectionable data-generating components from neutral ones, using reference points and a value instantiation rule.

**Strengths:**

- The paper studies a difficult, important, and often overlooked issue.
- The framework is grounded in a well-established philosophical theory of justice.
- I liked the idea of using reference points to decouple objectionable components in the data generation process of the predictive outcome.
- I also appreciated the comparison against existing approaches.

**Weaknesses:**

- It's unclear how broadly the framework can be applied across different domains and whether there are any limitations to its scalability. In particular, the evaluation seems to be limited, with the only "real-world" dataset used is the UCI adult dataset where  only 6 features are used.
- I also did not find a discussion regarding the practicality of implementing the proposed framework.

**Questions:**

1. Have you tried to use the proposed framework to more complex datasets (more variables and larger domains)? How does it scale?
What happens when you have multiple confounding variables?

2. As a follow-up from the previous question; What are the computational costs associated with implementing the framework, and how do they compare to existing methods?

---

> ### Author Response · Authors · 2023-11-20
> **Response to Comments by Reviewer 2K3E**
>
> We are very grateful for your constructive and insightful comments, and for the time and effort devoted! Below please see our responses to specific points in the review comment:
>
> ---
>
> ### **C1:** "It's unclear how broadly the framework can be applied across different domains and whether there are any limitations to its scalability."
>
> **R1:** Thanks for considering the applicability and scalability of our approach! Our framework can be applied to linear and nonlinear data generating processes, and scales well with the number of variables. To demonstrate the scalability of our framework, we have included a quantitative illustration of the computational cost (with more than 1,000 variables) in Appendix D.1.2. The computational cost of our approach is comparable to that of the vanilla regression model (implemented with a two-hidden-layer neural network). For the convenience of locating the related material, we provide the side note `Re: C1 by Reviewer 2K3E` on page 24.
>
> ---
>
> ### **C2:** "[The reviewer] did not ﬁnd a discussion regarding the practicality of implementing the proposed framework."
>
> **R2:** Thanks for carefully considering the practicality of implementing our framework! In addition to our code implementation in the supplementary material (we provided [the anonymous link](https://anonymous.4open.science/r/DecouplingObjectionable) in the paper), we have also included descriptions of our implementation details in Appendix D.1.1, along with the side note `Re: C2 by Reviewer 2K3E` on page 24.
>
> ---
>
> ### **Q3:** "Have you tried to use the proposed framework to more complex datasets (more variables and larger domains)?"
>
> **R3:** Thanks for the question on experiments with additional data. In addition to the two sets of experiments on simulated data and UCI Adult data set, we have included additional experimental results on the real-world Folktables data set (Ding et al., 2021). In particular, we consider the public health coverage prediction task where there are 17 features and 1 target variable. In light of the comment, we have included the additional experimental results in Appendix D.4, along with the side note `Re: Q3 by Reviewer 2K3E` on page 28.
>
> ---
>
> ### **Q4:** "What happens when you have multiple confounding variables?"
>
> **R4:** Thanks for the thoughtful question on confounding variables! We believe the term "confounding variables" in the question refers to the (potentially latent) confounders that connect with observed nodes via direct edges coming into the nodes. Our value instantiation rule (Algorithm 1) still works when there are confounders. Among the three options (Steps 5 -- 10 of Algorithm 1), the reference point value configuration focuses on the tail node of the objectionable components (edges going out of the tail node). The other two options, namely, the downstream of reference points and the original value in data, naturally adapt to the scenario with confounders, since we propagate the information from causal upstream to causal downstream. We provide the side note `Re: Q4 by Reviewer 2K3E` on page 6. Please kindly let us know if we misunderstood the question.
>
> ---
>
> ### **Q5:** "What are the computational costs associated with implementing the framework, and how do they compare to existing methods?"
>
> **R5:** Thanks for the question about computational costs. The decoupling of objectionable data generating components does not require retraining the modeling for local causal modules. The computational overhead only comes from the inference phase of the model during the simulated annealing process (to optimize reference point configurations). The computation cost of our approach is comparable to that of existing methods based on constrained optimization. In light of the question, we have included the quantitative illustration as well as the above discussion on computational costs in Appendix D.1.2, along with the side note `Re: Q5 by Reviewer 2K3E` on page 24.
>
> ---
>
> ### **Reference**
>
> Frances Ding, Moritz Hardt, John Miller, and Ludwig Schmidt. Retiring adult: New datasets for fair machine
> learning. In _Advances in Neural Information Processing Systems_, volume 34, pp. 6478–6490, 2021

---

> > ### Comment · Reviewer_2K3E · 2023-11-21
> > **Response**
> >
> > Thank you for the thoughtful replies to my questions. It is greatly appreciated.

---

> > > ### Author Response · Authors · 2023-11-21
> > > **Thank Reviewer for the Follow-Up**
> > >
> > > Thank you again for the constructive and insightful comments, which helped us improve our work! We greatly appreciate the time and effort devoted. Please kindly let us know if you would like to suggest any further changes.

---

### Meta-Review · Area_Chair_3eVk · 2023-12-12

**Metareview:**

This paper presents an interesting take on procedural fairness with an emphasis on causality.  The paper is strongly tied to maximin fairness -- something that's been taken on by the community (the ML community specifically motivated by Rawls' works and others outside of pure ML); all of the complaints around that style of fairness still hold here.  The paper also strongly relies on the ability to tease out causal models/connections, with many of the examples in the paper being relatively simple in that regard.  Still, the paper provides a well-written, creative, novel, and overall valuable constructive critique on this style of fair ML, and would be at home at ICLR (or any top ML conference).

**Justification For Why Not Higher Score:**

A good paper, a good fit for ICLR, but not a perfect paper - reviewers bring up many potential improvements.

**Justification For Why Not Lower Score:**

This would be a good spotlight (or even oral, were the reviews stronger).  It's an interesting paper, an interesting and nuanced take on a field that's full of incremental work, and the authors of the work bring a diverse perspective to the writing (and potential presentation).  Reviewers seems to agree.

---

### Decision · Program_Chairs · 2024-01-16

Accept (spotlight)